# Different RONS Generation in MTC-SK and NSCL Cells Lead to Varying Antitumoral Effects of Alpha-Ketoglutarate + 5-HMF

Joachim Greilberger [1,*] , Katharina Erlbacher [2], Philipp Stiegler [3], Reinhold Wintersteiger [4] and Ralf Herwig [5,6]

1   Institut für Laborwissenschaften Dr. Greilberger, Schwarzl Medical Center, 8301 Lassnitzhoehe, Austria
2   HG Pharma GmbH, 6365 Kirchberg in Tirol, Austria
3   Division of Transplantation Surgery, Medical University of Graz, 8010 Graz, Austria
4   Department of Pharmaceutical Chemistry, Institute of Pharmaceutical Sciences, University of Graz, 8010 Graz, Austria
5   Laboratories PD Dr. R. Herwig, 80337 Munich, Germany
6   Heimerer-College, 10000 Pristina, Kosovo
\*   Correspondence: joachim.greilberger@medunigraz.at

**Abstract:** Background: Carbonylated proteins (CPs) serve as specific indicators of increased reactive oxygen and nitrogen species (RONS) production in cancer cells, attributed to the dysregulated mitochondrial energy metabolism known as the Warburg effect. The aim of this study was to investigate the potential of alpha-ketoglutarate (aKG), 5-hydroxymethylfurfural (5-HMF), and their combination as mitochondrial-targeting antioxidants in MTC-SK or NCI-H23 cancer cells. Methods: MTC-SK and NCI-H23 cells were cultured in the absence or presence of varying concentrations (0–500 μg/mL) of aKG, 5-HMF, and the combined aKG + 5-HMF solutions. After 0, 24, 48, and 72 h, mitochondrial activity, cancer cell membrane CP levels, cell growth, and caspase-3 activity were assessed in aliquots of MTC-SK and NCI-H23 cells. Results: The mitochondrial activity of MTC-SK cells exhibited a concentration- and time-dependent reduction upon treatment with aKG, 5-HMF, or the combined aKG + 5-HMF. The half-maximal inhibitory concentration (IC50%) for mitochondrial activity was achieved at 500 μg/mL aKG, 200 μg/mL 5-HMF, and 200 μg/mL aKG + 66.7 μg/mL 5-HMF after 72 h. In contrast, NCI-H23 cells showed a minimal reduction (10%) in mitochondrial activity even at the highest combined concentration of aKG + 5-HMF. The CP levels in MTC-SK cells were measured at 8.7 nmol/mg protein, while NCI-H23 cells exhibited CP levels of 1.4 nmol/mg protein. The combination of aKG + 5-HMF led to a decrease in CP levels specifically in MTC-SK cells. The correlation between mitochondrial activity and CP levels in the presence of different concentrations of combined aKG + 5-HMF in MTC-SK cells demonstrated a linear and concentration-dependent decline in CP levels and mitochondrial activity. Conversely, the effect was less pronounced in NCI-H23 cells. Cell growth of MTC-CK cells was reduced to 60% after 48 h and maintained at 50% after 72 h incubation when treated with 500 μg/mL aKG (IC50%). Addition of 500 μg/mL 5-HMF inhibited cell growth completely regardless of the incubation time. The IC50% for 5-HMF on MTC-CK cell growth was calculated at 375 μg/mL after 24 h incubation and 200 μg/mL 5-HMF after 72 h. MTC-SK cells treated with 500 μg/mL aKG + 167 μg/mL 5-HMF showed no cell growth. The calculated IC50% for the combined substances was 250 μg/mL aKG + 83.3 μg/mL 5-HMF (48 h incubation) and 200 μg/mL aKG + 66.7 μg/mL 5-HMF (72 h incubation). None of the tested concentrations of aKG, 5-HMF, or the combined solution had any effect on NCI-H23 cell growth at any incubation time. Caspase-3 activity increased to 21% in MTC-CK cells in the presence of 500 μg/mL aKG, while an increase to 59.6% was observed using 500 μg/mL 5-HMF. The combination of 500 μg/mL aKG + 167.7 μg/mL 5-HMF resulted in a caspase-3 activity of 55.2%. No caspase-3 activation was observed in NCI-H23 cells when treated with aKG, 5-HMF, or the combined solutions. Conclusion: CPs may serve as potential markers for distinguishing between cancer cells regulated by RONS. The combination of aKG + 5-HMF showed induced cell death in high-RONS-generating cancer cells compared to low-RONS-generating cancer cells.

**Keywords:** medullary thyroid cancer cell (MTS-SK); non-small-cell lung carcinoma (NSCLC); alpha-ketoglutarate (aKG); 5-hydroxy-methyl-furfural (5-HMF); carbonylated proteins of cell membrane (CP); reactive oxygen and nitrogen species (RONS); cell proliferation; mitochondrial activity; caspase activity

## 1. Introduction

Plasma protein carbonylation is involved in various types of cancer, such as hematological malignancies, colorectal cancer, gastric cancer, and medullary thyroid carcinoma [1]. However, little is known about the levels of carbonylated proteins in cancer cell tissue compared to healthy cells. Specific carbonylated proteins are found in higher amounts in human breast cancer compared to adjacent healthy epithelial tissue, due to low superoxide dismutase activities [2]. In normal cells, the activity of mitochondrial electron transport is directly proportional to reactive oxygen species (ROS) production, including superoxide anion radicals ($O_2^{\bullet-}$), whereas, in cancer cells, the higher ROS levels are primarily dependent on the mitochondrial source [3]. This is due to the mutation of isocitrate dehydrogenase and the higher expression of NADPH-oxidases (NOX), which catalyze the generation of $O_2^{\bullet-}$ in various locations, including the membrane, mitochondria, and nucleus, leading to the formation of carbonylated proteins. The content of carbonylated proteins may serve as a stable marker of ROS in the dysregulated mitochondrial energy synthesis of cancer cells, known as the Warburg effect [4]. While compounds such as alpha-lipoic acid and hydroxycitrate have been shown to slow cancer growth in various tumors, additional strategies are needed to correct the metabolic bottleneck in energy generation and target mitochondrial activity, which are intertwined phenomena in the Warburg effect.

In recent years, alpha-ketoglutarate (aKG), an endogenous intermediate metabolite of the Krebs cycle, has gained attention as a potential novel anticancer agent [5]. Alpha-ketoglutarate is involved in the activation of $Fe^{2+}$, $O_2$, and ascorbate-dependent 2-oxoglutarate-dependent dioxygenases (2-OGDDs), which play a role in epigenetic regulation, such as DNA demethylation by Jumonji C domain-containing lysine demethylases (KDM 2–7) and translocation hydroxylases (TET 1–3) [6–8]. These epigenetic pathways inhibit carcinogenesis in the presence of aKG but are blocked by specific oncometabolites such as succinate, fumarate, and 2-hydroxyglutarate. The accumulation of succinate occurs due to additional oxidative damage of aKG during excessive radical formation, specifically hydrogen peroxides or peroxynitrite. Additionally, aKG is involved in non-epigenetic inhibitory processes within the 2-OGGDs pathways, regulating prolyl hydroxylase domain-containing protein 2 (PHD2), which is responsible for the inactivation of hypoxia-inducible factor alpha (HIF$\alpha$). Inhibition of HIF$\alpha$ by hypoxia and/or oncometabolites such as succinate promotes carcinogenesis and tumor progression [9]. In cell studies, supplementation of aKG has demonstrated antitumoral potential in osteosarcoma cell lines (Saos-2 and HOS), inducing concentration-dependent effects [5] such as TP53 mutation-dependent antitumor effects in osteosarcoma cells and aKG-induced apoptosis via caspase 3 activity in leukemic cells [10]. The combination of aKG and 5-hydroxymethyl-furfural (5-HMF) has been shown to potentiate the inhibitory effect of aKG, leading to a total breakdown of cell growth and synergistic induction of apoptosis involving both enzymatic and nonenzymatic processes. In healthy cell lines, such as human skin fibroblasts, aKG, 5-HMF, and their combination do not affect cell growth or apoptosis.

In contrast to normal cells, where high oxidative stress levels lead to apoptosis, cancer cells take an advantage of an oxidative milieu. Cancer cells build up a complex environment including stroma components which enhance oxidative stress, resulting in tumor growth. Experimental and clinical trials showed the key role of oxidative stress in several tumor aspects, thus influencing the characteristics of tumor cells. Intracellular pathways and the potential of DNA mutations that control the cell proliferation, the cell survival, the cell motility, and cell invasiveness, as well as the stromal components, are affected by

oxidants. A dysregulation in these pathways strongly affects cancer development, cancer dissemination, inflammation, tissue repair, and neo-angiogenesis [11].

A new study from Hanahan and Weinberg defined two new "hallmarks of cancer" depending on oxidative stress: the tumor microenvironment and the metabolic reprogramming of cancer cells. Intratumoral hypoxia and/or infiltration of CAFs should be treated with a disruption of the Warburg metabolism in cancer and stromal cells, as well as a reconfiguration through the antioxidative strategy of the pentose phosphate pathway. Pharmacological approaches could be medicaments which affect the lactate shuttle and inhibitors of glycolysis in combination with inhibitors of autophagy. Autophagy is a compensation mechanism of nutrient-poor cancer cells. In further studies, the effects of all biochemical reactions of ROS in cancer cells should be investigated [12].

Medullary thyroid carcinoma (MTC) cells, which are neoplasms of the endocrine system, account for 10% of all thyroid cancers and are mainly caused by mutations in the RET proto-oncogene from para-follicular C-cells. Oxidative stress factors, such as carbonyl groups (malondialdehyde, MDA), and reduced antioxidative regulating enzymes have been associated with the risk of MTC in patients [13]. The detection of carbonylated proteins in membrane proteins isolated from leukemia cell lines may indicate a higher turnover of oxidative potential in misdirected mitochondrial energy metabolism. It is speculated that aKG and 5-HMF induce apoptosis in Jurkat cells by reducing reactive oxygen and nitrogen species (RONS). Normal fibroblasts with low RONS levels remain stable due to the normal metabolism of aKG as an endogenous substance and 5-HMF as a carbohydrate [10].

Non-small-cell lung cancer (NSCLC) has a high incidence rate and is often resistant to chemotherapy and radiotherapy. Pro-oxidative treatment of NCI-H23 cells with vincenin-2, a pro-oxidant flavonoid, has been shown to increase carbonylated proteins and activate apoptotic genes, leading to cell death. Additional radiation further increases carbonylated proteins in NCI-H23 cells, promoting cell death [14]. Pachymic acid has also exhibited antitumor effects by inducing reactive oxygen species and activating c-Jun N-terminal kinase (JNK) and endoplasmic reticulum stress apoptotic pathways in lung cancer cells [15]. It appears that only pro-oxidative reactions can increase cell death in NCI-H23 cells, including radiation, whereas anti-oxidative substances do not have the same effect.

The aim of this study was to investigate the antiproliferative and antitumoral effects of aKG and 5-HMF, which are associated with reduced mitochondrial activity, increased apoptosis, and a decrease in oxidative stress factors such as carbonylated membrane proteins in MTC-SK and NCI-H23 cells.

## 2. Materials and Methods

### 2.1. Materials

The following materials were used: media—RPMI-1640, FBS, DMEM, and antibiotic–antimycotic solution (Thermofisher, Vienna, Austria); reagents—Cytofix–Cytoperm permeabilization Kit (Thermofisher, Vienna, Austria), FITC active caspase-3 apoptosis kit (BD Biosciences Kit; Allschwil, Germany), and WST-1 cell proliferation reagent (Abcam; Cambridge, UK); chemicals—alpha-ketoglutarate (Sigma, Vienna, Austria), 5-hydroxymethyl-furfurale (5-HMF, Evonik Operation, Darmstadt, Germany), guanidine-HCl, butyl-hydroxy-toluene (BHT; Sigma Aldrich; Vienna, Austria), and di-nitro-phenyl-hydrazine (DNPH) (Altmann Analytics, Munich, Germany); flasks (Falco® Cell Culture; Corning Incorporated, 14831 New York, NY, USA).

### 2.2. Cell Culture

The medullary thyroid carcinoma cells (MTC-SK) were provided by Prof. Dr. Pfragner at the Medical University of Graz. In the MTC-SK cell line, the predominant findings were terminal chromosomal rearrangements most frequently concerning chromosome 11p, i.e., the locus of the calcitonin and calcitonin gene-related peptide genes and the H-ras oncogene, and a characteristic instability of the centromeric region of chromosome 16 and somatic pairing of the homologous chromosomes 16 [16]. The MTC-SK cells were resus-

pended in RPMI-1640 with glutamine, 1% antibiotic–antimycotic, and 10% FBS, and the incubation density for further experiments was approximately 1,000,000 cells per mL. The human lung cancer cell line NCI-H23 was purchased from the American Type Culture Collection (Rockville, MD, USA), and then cultured as recommended as monolayers in DMEM medium supplemented with 10% fetal bovine serum and 1% penicillin–streptomycin–neomycin in a humidified incubator at 37 °C in a 5% $CO_2$ atmosphere.

### 2.3. Cell Proliferation Experiments

In line with our previous study [10], we used different concentrations of aKG (0–500 μg/mL), 5-HMF (0–500 μg/mL), or aKG + 5-HMF s (0 μg aKG + 0 μg/mL 5-HMF, 125 μg aKG + 41.7 5-HMF μg/mL, 200 μg aKG + 66.7 μg 5-HMF μg/mL, 250 μg AKG + 83.3 μg/mL 5-HMF, 375 μg aKG + 125.4 μg/mL 5-HMF, and 500 μg aKG + 166.7 μg/mL 5-HMF) in medium for 24, 48, and 72 h to investigate the cell proliferation. The cell growth was determined using the CASY® Cell Counter (Hoffmann-La Roche Ltd., Basel, Switzerland). Aliquots were taken for further experiments.

### 2.4. Detection of the Mitochondrial Membrane Potential with Flow Cytometry

A population of $2.0 \times 10^6$ cells/mL was centrifuged ($400 \times g$) after incubation for 72 h without any substances (negative control), with 4 or 0.4 μM CPT (positive control), and with 250 μg/mL aKG + 83.3 μg/mL 5-HMF or 500 μg/mL aKG + 166.7 μg/mL 5-HMF. The cells were resuspended in 0.5 mL of JC-1 solution (5,5′,6,6′-tetracholor-1,1′,3,3′-tertaehtylbenzimidazolcarbocyanin-iodide; 125 μL of stock solution + 12.375 mL of assay buffer) and incubated for 15 min with 5% $CO_2$ at 37 °C, followed by two washing steps using the assay buffer (BD Biosciences Kit; 4123 Allschwil, Germany). Afterward, the cells were resuspended in 1 mL of assay buffer and analyzed by FACS. The aggregate dye produced the extinction of the monomer at 535 nm and the aggregate at 475 nm.

### 2.5. Caspase-3 Activity Measurements

Caspase-3-activated apoptosis was investigated as described in our previous study [10]. The cells were incubated for 72 h in absence or presence of 1.7 or 3.5 μM of aKG, 2 or 4 μM 5-HMF, 1.7 μg/mL aKG + 0.7 μM 5-HMF, and 3.5 μM aKG + 1.3 μM 5-HMF, and subsequently separated by centrifugation at $3500 \times g$ for 10 min. Afterward, the cells were washed for several times using cold PBS, and 500 μL of Cytofix–Cytoperm was added on ice at −20 °C for 20 min. After incubation, the cells were labeled with FITC-tagged antibodies, and the examination of the caspase activity at 495 nm was performed using the washing step.

### 2.6. Cell Lysate Protein Damage Measurements

The cell lysate's protein concentrations were estimated using the BSA test kit. For triple estimations, 25 μL of diluted albumin standards and cell lysates were transferred into microtitration plates. Afterward, BCA reagent solution (50:1) was added, and the microtitration plate was closed using a top cover. The plates were incubated for 30 min using a plate shaker at 37 °C and 5% $CO_2$. The calculation of the protein concentrations was performed at 562 nm (Spectra Max Photometer Pro 384) as soon as the plates reached room temperature. Furthermore, to calculate (investigate, measure) the carbonylated proteins, the samples were diluted to 4 mg/mL protein. About 1 mg of protein was precipitated by pipetting 250 μL of the 4 mg/mL protein solution with 250 μL of 20% trichloric acid (TCA) solution. All precipitated samples (1 mg) were centrifuged at $5000 \times g$ for 3 min, and the supernatant was removed. Pellets (1 mg) were dissolved in 10 mM DNPH containing 6 M guanidine pH 2.5 and incubated for 45 min at room temperature. Protein damage was measured spectrophotometrically with the protein carbonyl assay [10]. Protein carbonyl data are expressed in nmol/mg membrane protein.

*2.7. Statistical Analysis*

Group comparisons were made using *t*-tests where appropriate and indicated. Linear regression and exponential regression curves were calculated on the basis of Pearson regression (SPSS 25, SPSS Inc., Chicago, IL, USA). All values are given as the mean and standard deviation. Statistical significance was set at $p < 0.05$, with high significance considered at $p < 0.01$.

## 3. Results

*3.1. Cell Proliferation Experiments*

aKG: Figure 1A shows the MTC-SK cell growth in absence or presence of different aKG concentrations (0, 125, 200, 250, 375, and 500 µg/mL) over 3 days compared to the control. After 24 h, no inhibiting effect of aKG was detected compared to the control. The cell growth after 48 h using different concentrations of aKG was significantly reduced with 250 µg/mL (38,145 ± 2835 cells, $n = 5$; $p < 0.01$), 375 µg/mL aKG (36,575 ± 1795 cells, $n = 5$; $p < 0.01$), and 500 µg/mL aKG (32,097 ± 2485 cells, $n = 5$; $p < 0.01$) compared to the control (0 µg/mL aKG: 45,183 ± 1345 cells, $n = 5$). There was a 29% reduction in proliferation using 500 µg/mL aKG compared to the control signal, and it was also significant lower compared to 375 µg/mL aKG. After 72 h, the inhibiting effect of aKG on the cell growth became more effective. While the usage of 125 µg/mL aKG showed no significant effect (63,088 ± 2213 cells, $n = 5$) compared to the control (67,521 ± 1178 cells, $n = 5$), 500 µg/mL aKG showed 37.2% inhibition (42,441 ± 1178 cells, $n = 5$; $p < 0.01$), followed by 375 µg/mL aKG (49,223 ± 2158 cells, $n = 5$; $p < 0.01$), 250 µg/mL aKG (55,342 ± 3685 cells, $n = 5$; $p < 0.01$), and 200 µg/mL aKG (59,211 ± 4571 cells, $n = 5$; $p < 0.01$). Linear regression (Figure 1A) showed a reduction in the linear slope depending on the dosage of aKG: The linear regression using 500 µg/mL aKG was more than twofold smaller (7.42) compared to the linear regression in absence of aKG, 40% smaller using 375 µg/mL aKG ($k = 9.84$), 28% smaller using 250 µg/mL aKG ($k = 11.79$), and 10% smaller using 200 µg/mL aKG ($k = 14.54$). The calculated IC50% of aKG was 570 µg/mL after 72 h incubation.

The cell growth of NCI-H23 cells was not influenced positively or negatively by aKG at any concentration because no significance was generated (Figure 1B).

5-HMF: Figure 2A shows the MTC-SK cell growth in absence or presence of different 5-HMF (0, 125, 200, 250, 375, and 500 µg/mL) concentrations over 3 days compared to the control. No significant difference in any 5-HMF concentrations was observed between 0 and 24 h incubation, as well as for both controls. The cell growth of MTC-SK cells started after 48 h. The usage of 250 µg/mL 5-HMF reduced the cell growth significantly compared to the control after 48 h (26,098 ± 5095 cells vs. 36,811 ± 5122; $n = 5$; $p < 0.05$), with the concentrations of 375 µg/mL reducing growth to 23,234 ± 3197 cells (37.5% reduction; $n = 5$; $p < 0.01$) and 500 µg/mL reducing growth to 21,023 ± 1543 cells (43% reduction; $n = 5$; $p < 0.01$). After 72 h, the inhibitory effect of 5-HMF on cell proliferation was generally more effective and resulted in a decreased cell growth of MTC-SK cells using different 5-HMF concentrations; a 38% reduction was measured using the lowest 5-HMF concentration of 125 µg/mL (48,109 ± 3876 cells; $n = 5$; $p < 0.01$) compared to the control (78,145 ± 4122 cells; $n = 5$), followed by 200 µg/mL 5-HMF (50% reduction; 38,624 ± 3342 cells; $n = 5$; $p < 0.01$), 250 µg/mL 5-HMF (56% reduction; 34,199 ± 3099 cells; $n = 5$; $p < 0.01$), 375 µg/mL 5-HMF (65% reduction; 27,488 ± 4144 cells; $n = 5$; $p < 0.01$), and 500 µg/mL 5-HMF (76% reduction; 18,742 ± 3222 cells; $n = 5$; $p < 0.01$). Correlation of the cell growth with 5-HMF concentrations during incubation was obtained for 0, 125, 200, and 250 µg/mL 5-HMF with an exponential increase. The usage of 375 and 500 µg/mL 5-HMF caused no cell growth. The calculated IC50% of 5-HMF was 200 µg/mL after 72 h incubation.

On NHI-H23 5-HMF showed no significant effect on cell growth using several concentrations (Figure 2B).

Combination of aKG + 5-HMF: Figure 3A describes the MTC-SK cell growth in the absence or presence of different combined aKG + 5-HMF (3:1) concentrations (0 µg/mL aKG + 0 µg/mL 5-HMF, 125 µg/mL aKG + 41.7 µg/mL 5-HMF, 200 µg/mL aKG +

66.7 µg/mL 5-HMF, 250 µg/mL aKG + 83.3 µg/mL 5-HMF, 375 µg/mL aKG + 125.4 µg/mL 5-HMF, and 500 µg/mL aKG + 166.7 µg/mL 5-HMF) over 3 days compared to the control. After 24 h incubation, the highest concentrated aKG + 5-HMF solution (500 µg/mL aKG + 166.7 µg/mL 5-HMF: 22,823 ± 1854 cells; $n$ = 5; $p < 0.01$) showed a significant reduction in cell proliferation for MTC-SK cells compared to the control (32,145 ± 2324 cells), and there was no significant difference compared to the control signal (0 h; 20,000 ± 2108 cells).

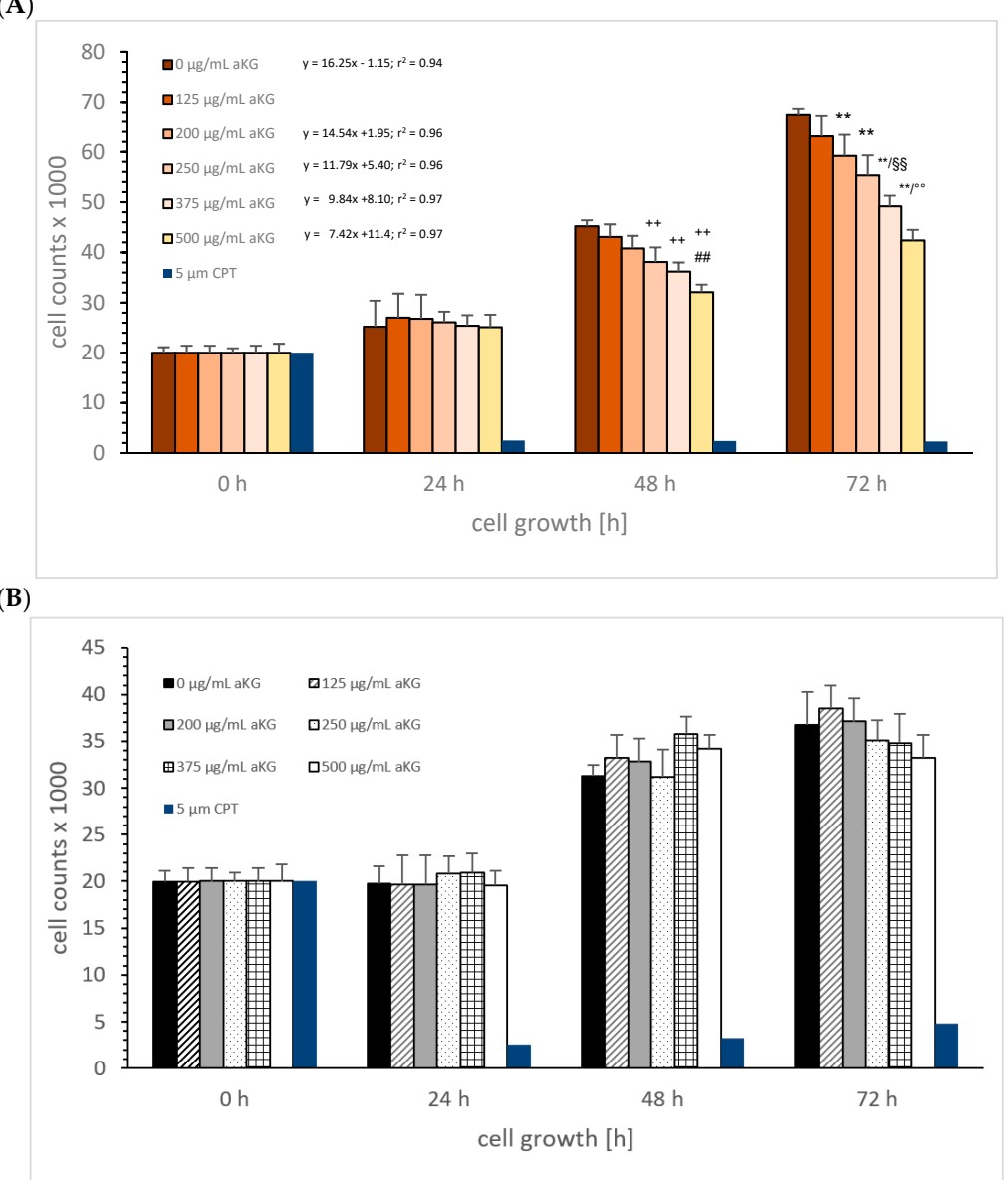

**Figure 1.** Cell growth of MTC-SK (**A**) and NCI-H23 (**B**) cells in the presence or absence of 0, 125, 200, 250, 375, and 500 µg/mL aKG, and linear regression of cell growth over time in absence or presence of aKG. MTC-SK: ** $p < 0.01$: significant differences after 72 h observed for 200, 250, 375, and 500 µg/mL aKG compared to the control (0 µg/mL aKG); §§ $p < 0.01$: significant differences after 72 h observed between 250 and 375 µg/mL aKG; °° $p < 0.01$: significant differences after 72 h observed between 375 and 500 µg/mL aKG; ++ $p < 0.01$: significant differences after 48 h observed for 250, 375, and 500 µg/mL aKG compared to the control (0 µg/mL aKG); ## $p < 0.01$: significant differences after 48 h observed between 375 and 500 µg/mL aKG.

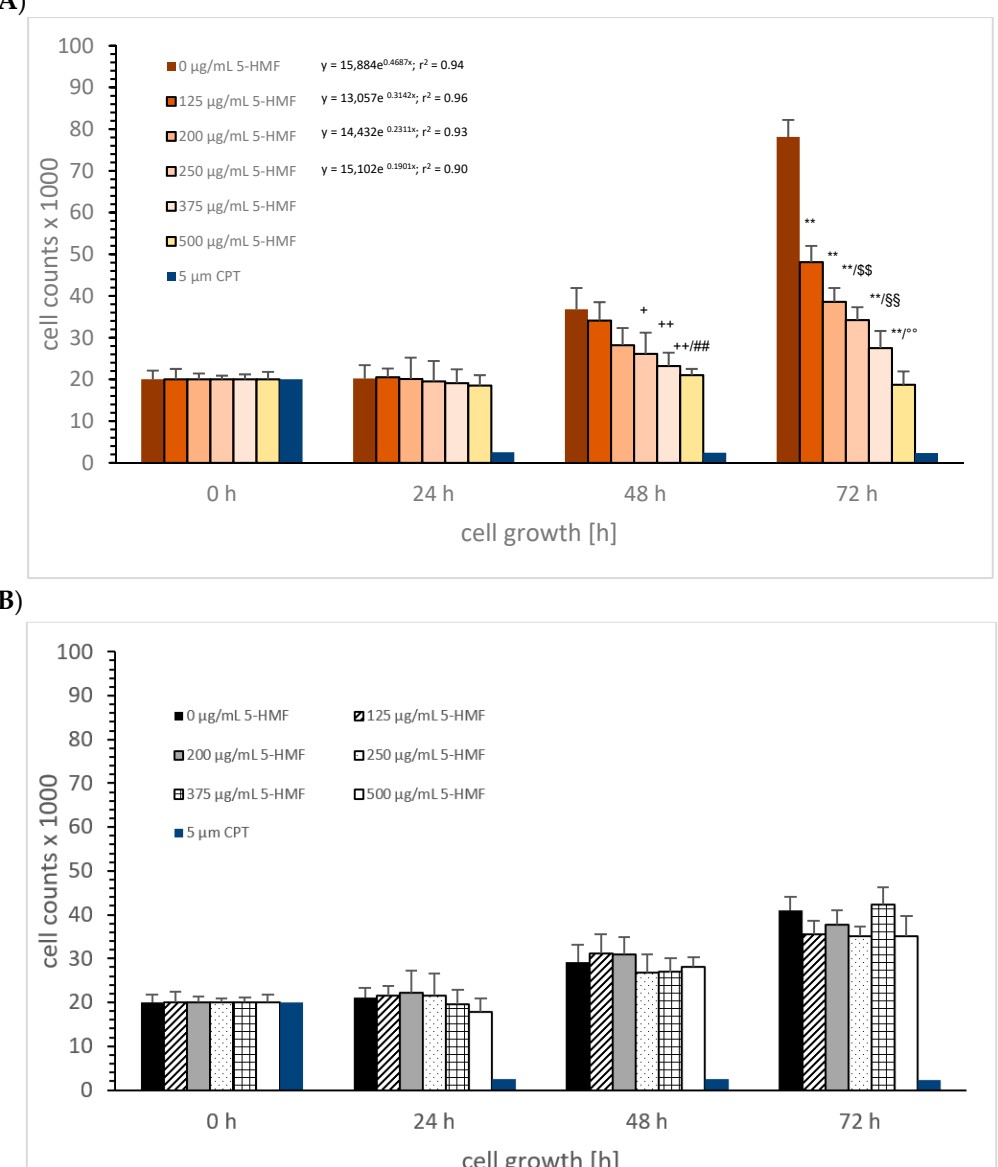

**Figure 2.** Cell growth of MTC-SK (**A**) and NCI-H23 (**B**) cells in presence or absence of 0, 125, 200, 250, 375, and 500 μg/mL 5-HMF and correlation with cell growth over time in absence or presence of 5-HMF. MTC-SK: $^{\$\$}$ $p < 0.01$: significant differences after 72 h observed between 125 and 200 μg/mL 5-HMF; $^{\S\S}$ $p < 0.01$: significant differences after 72 h observed between 200 and 250 μg/mL 5-HMF; $^{\circ\circ}$ $p < 0.01$: significant differences after 72 h observed between 375 and 500 μg/mL 5-HMF. $^{+}$ $p < 0.05$: significant differences after 48 h observed between 0 and 250 μg/mL 5-HMF; $^{++}$ $p < 0.01$: significant differences after 48 h observed between 0 and 250, 375 and 500 μg/mL 5-HMF; $^{\#\#}$ $p < 0.05$: significant differences after 48 h observed between 200 and 250 μg/mL 5-HMF; $^{**}$ $p < 0.01$: significant differences after 72 h observed for 125, 200, 250, 375, and 500 μg/mL 5-HMF compared to the control (0 μM 5-HMF).

**(A)**

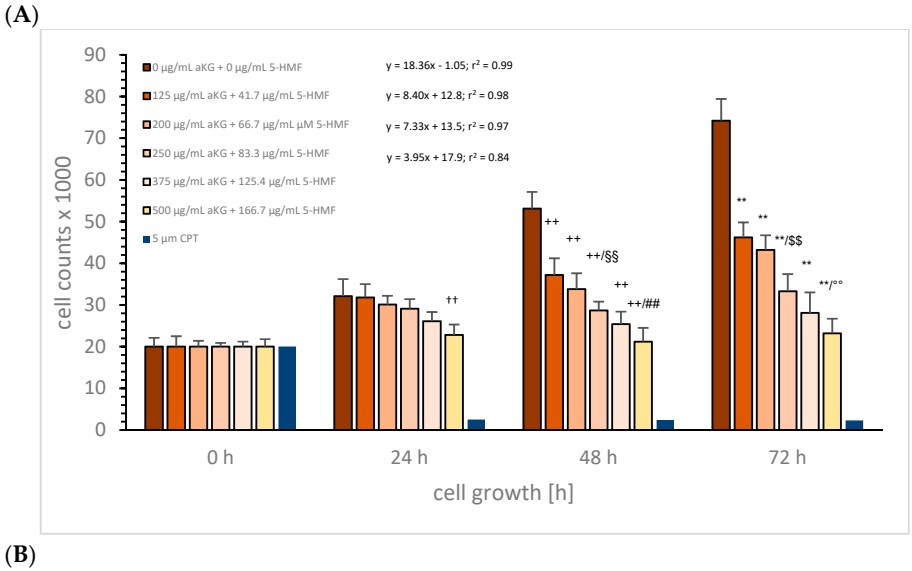

**(B)**

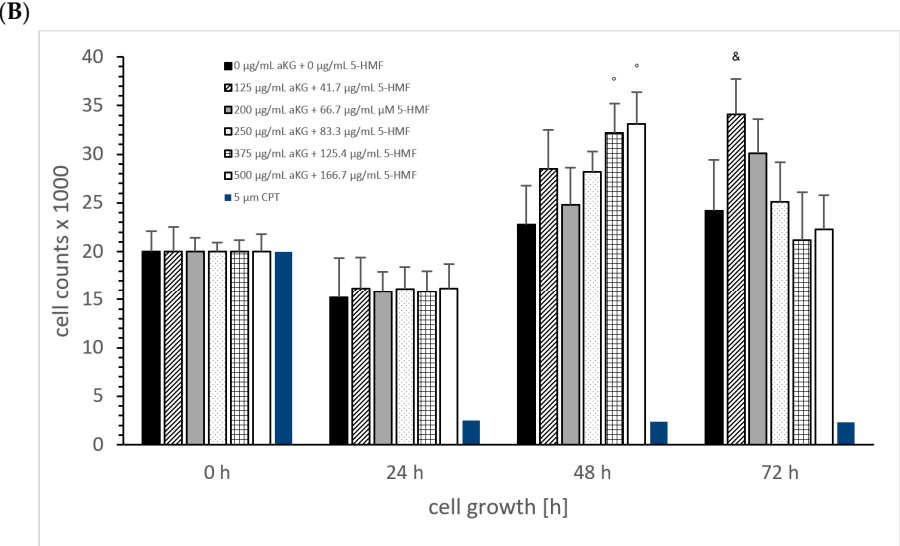

**Figure 3.** Cell growth of MTC-SK (**A**) and NCI-H23 (**B**) cells in presence or absence of combined aKG + 5-HMF solutions and correlation of the cell growth over time in absence or presence of aKG + 5-HMF. MTC-SK: ** $p < 0.01$: significant differences after 72 h observed for 125 µg/mL aKG + 41.7 µg/mL 5-HMF, 200 µg/mL aKG + 66.7 µg/mL 5-HMF, 250 µg/mL aKG + 83.3 µg/mL 5-HMF, 375 µg/mL aKG + 125.4 µg/mL 5-HMF, and 500 µg/mL aKG + 166.7 µg/mL 5-HMF compared to the control (0 µg/mL aKG + 0 µg/mL 5-HMF); $^{\$\$}$ $p < 0.01$: significant differences after 72 h observed between 200 µg/mL aKG + 66.7 µg/mL 5-HMF and 250 µg/mL aKG + 83.3 µg/mL 5-HMF; °° $p < 0.01$: significant differences after 72 h observed between 375 µg/mL aKG + 125.4 µg/mL 5-HMF and 500 µg/mL aKG and 500 µg/mL aKG + 166.7 µg/mL 5-HMF; $^{++}$ $p < 0.01$: significant differences after 48 h observed for 125 µg/mL aKG + 41.7 µg/mL 5-HMF, 200 µg/mL aKG + 66.7 µg/mL 5-HMF, 250 µg/mL aKG + 83.3 µg/mL 5-HMF, and 500 µg/mL aKG + 166.7 µg/mL 5-HMF compared to the control (0 µg/mL aKG + 0 µg/mL 5-HMF); $^{\#\#}$ $p < 0.01$: significant differences after 48 h observed between 250 µg/mL aKG + 83.3 µg/mL 5-HMF and 500 µg/mL aKG + 166.7 µg/mL 5-HMF; $^{\S\S}$ $p < 0.01$: significant differences after 48 h observed between 125 µg/mL aKG + 41.7 µg/mL 5-HMF and 250 µg/mL aKG + 83.3 µg/mL 5-HMF; $^{\dagger\dagger}$ $p < 0.01$: significant differences after 24 h observed between 200 µg/mL aKG + 66.7 µg/mL 5-HMF and 500 µg/mL aKG + 166.7 µg/mL 5-HMF. NCI-H23: ° $p < 0.05$: significant differences after 48 h observed between control and 375 µg/mL aKG + 125.4 µg/mL 5-HMF and 500 µg/mL aKG or 500 µg/mL aKG + 166.7 µg/mL 5-HMF; $^{\&}$ $p < 0.05$: observed significant differences after 48 h between control and 125 µg/mL aKG + 41.7 µg/mL 5-HMF and 500 µg/mL aKG.

Each combined aKG + 5-HMF solution showed a significant reduction in the MTC-SK cell growth after 48 h. The control (53,143 ± 4311 cells) showed a significantly higher cell growth compared to 125 µg/mL aKG + 41.7 µg/mL 5-HMF (37,234 ± 4025 cells; $n = 5$; $p < 0.01$) with a reduction of nearly 30%, to 200 µg/mL aKG + 66.7 µg/mL 5-HMF (33,786 ± 3843 cells; $n = 5$; $p < 0.01$) with a reduction of 47%, to 250 µg/mL aKG + 83.3 µg/mL 5-HMF (28,652 ± 2177 cells; $n = 5$; $p < 0.01$) with a reduction of 47%, to 375 µg/mL aKG + 125.4 µg/mL 5-HMF (25,465 ± 2967 cells; $n = 5$; $p < 0.01$) with a reduction of 53%, and to 500 µg/mL aKG + 166.7 µg/mL 5-HMF (21,200 ± 3325 cells; $n = 5$; $p < 0.01$) with a reduction of 60%. No difference in cell growth was detected between the highest concentration of the combined solution after 48 h and the starting conditions.

After 72 h incubation, we obtained similar results compared to 48 h incubation in a concentration-dependent manner. The highest reduction in cell growth, 69%, was estimated using 500 µg/mL aKG + 166.7 µg/mL 5-HMF (23,178 ± 3472 cells; $n = 5$; $p < 0.01$) compared to the control (74,233 ± 5846 cells; $n = 5$), followed by 375 µg/mL aKG + 125.4 µg/mL 5-HMF (62% reduction; 28,096 ± 4869 cells; $n = 5$; $p < 0.01$), 250 µg/mL aKG + 83.3 µg/mL (55% reduction; 33,321 ± 4105 cells; $n = 5$; $p < 0.01$), 200 µg/mL aKG + 66.7 µg/mL 5-HMF (42% reduction; 43,214 ± 3511 cells; $n = 5$; $p < 0.01$), and 125 µg/mL aKG + 41.7 µg/mL 5-HMF (36% reduction; 46,239 ± 3558 cells; $n = 5$; $p < 0.01$).

The linear regression showed a correlation between the cell growth over time and the used concentration of the combined substances aKG + 5-HMF. While no significant increase was measured within 72 h incubation using 500 µg/mL aKG + 166.7 µg/mL 5-HMF, or 375 µg/mL aKG + 125.4 µg/mL 5-HMF, the slope of linear regression was highest during incubation with the lowest concentration (125 µg/mL aKG + 41.7 µg/mL 5-HMF; $k = 8.40$), followed by 200 µg/mL aKG + 66.7 µg/mL 5-HMF aKG + 0.52 µM 5-HMF ($k = 7.33$), and 250 µg/mL aKG + 83.3 µg/mL 5-HMF ($k = 3.95$). The normal cell growth of MTC-SK cells showed a linear regression of 18.4, which was more than four-fold higher than with 250 µg/mL aKG + 83.3 µg/mL 5-HMF. Additionally, the IC50% was calculated for the combined 250 µg/mL aKG + 83.3 µg/mL 5-HMF solution, resulting in a lower concentration compared to the IC50% of the single components, e.g., aKG (500 µg/mL) and 5-HMF (250 µg/mL).

Using NCI-H23 cells, aKG + 5-HMF showed less effects on cell growth. After 48 h, the combined solutions with concentrations of 375 µg/mL aKG + 125.4 µg/mL 5-HMF (32.2 ± 3.0%; $n = 5$; $p < 0.05$) and 500 µg/mL aKG + 166.7 µg/mL 5-HMF (33.1 ± 3.3%; $n = 5$; $p < 0.05$) significantly increased the cell growth of NCI-H23 compared to the control (22.8 ± 4.0%; $n = 5$; $p < 0.05$). The same was detected after 72 h with 125 µg/mL aKG + 41.7 µg/mL 5-HMF (34.1 ± 3.6%; $n = 5$; $p < 0.05$) compared to the control (24.2 ± 5.2%; $n = 5$; $p < 0.05$). Interestingly, highest concentrations of aKG + 5-HMF showed no cell proliferation compared to starting conditions.

### 3.2. Cytotoxic Assay

aKG: Aliquots of the cell growth experiments were used to measure mitochondrial activity. Figure 4A shows the mitochondrial activity of MTC-SK cells in absence or presence of different aKG concentrations at 0, 24, 48, and 72 h. After 24 h incubation with 200, 250, 375, and 500 µg/mL aKG, there was a significant decrease in the mitochondrial activity of MTC-SK cells compared to the control (100 ± 3.1%; $n = 5$); using 200 µg/mL aKG the mitochondrial activity decreased to 83.2 ± 3.1% ($n = 5$; $p < 0.01$), while the concentration of 250 µg/mL aKG led to a decrease to 70.0 ± 3.2%, ($n = 5$; $p < 0.01$), 375 µg/mL aKG led to a decrease to 61.2 ± 2.1%, ($n = 5$; $p < 0.01$), and 500 µg/mL aKG led to a decrease to 50.1 ± 1.8% ($n = 5$; $p < 0.01$). The lowest concentration of 125 µg/mL aKG (102.3 ± 4.0%; $n = 5$) had no effect on the mitochondrial activity of MTC-SK cells.

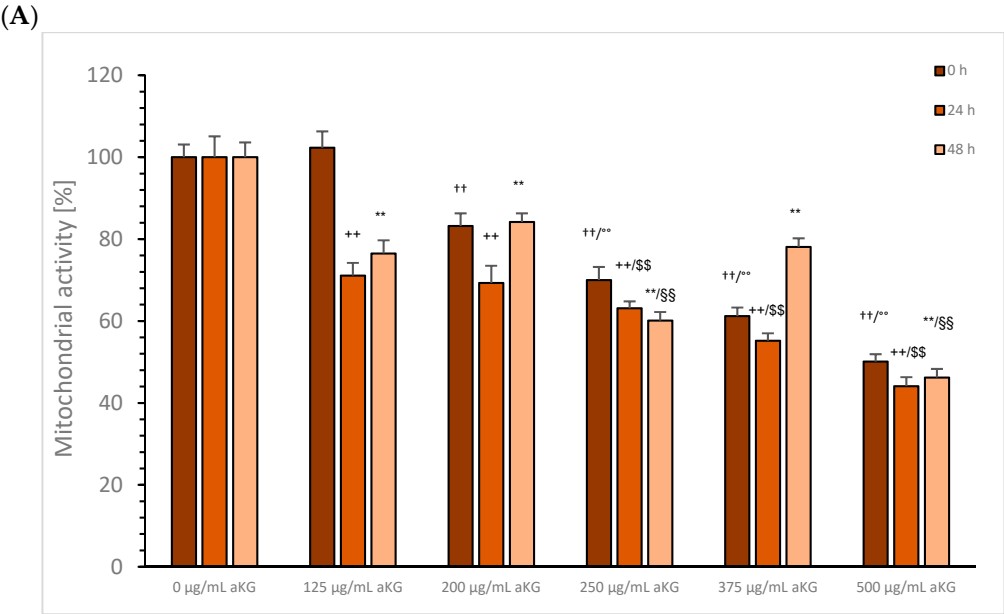

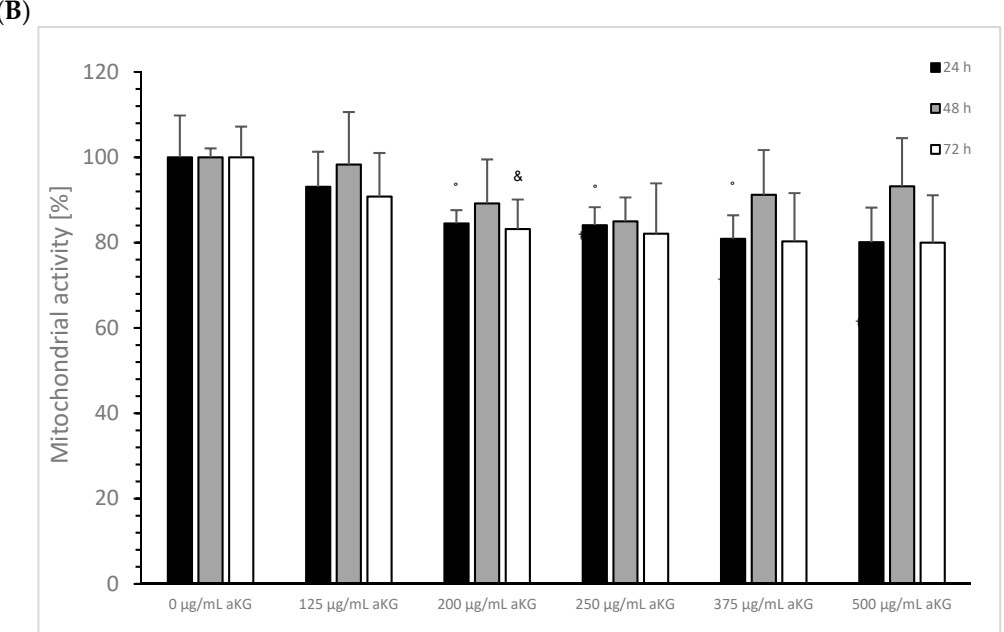

**Figure 4.** Mitochondrial activity of MTC-SK (**A**) and NCI-H23 (**B**) cells in presence or absence of 0, 125, 200, 250, 375, and 500 µg/mL aKG after 24, 48, and 72 h incubation. MTC-SK: $^{\dagger\dagger}$ $p < 0.01$: significant differences after 24 h observed for 200, 250, 375, and 500 µg/mL aKG compared to the control (0 µg/mL aKG); $^{\circ\circ}$ $p < 0.01$: significant differences observed between 200 and 250 µg/mL aKG, between 250 and 375 µM, and between 375 and 500 µg/mL aKG. $^{++}$ $p < 0.01$: significant differences after 48 h observed for 125, 200, 250, 375, and 500 µg/mL aKG compared to the control; $^{\$\$}$ $p < 0.01$: significant differences after 48 h observed between 125 and 250 µg/mL aKG, between 250 and 375 µg/mL aKG, and between 375 and 500 µg/mL aKG; ** $p < 0.01$: significant differences after 72 h observed for 200, 250, 375, and 500 µg/mL aKG compared to the control; $^{\S\S}$ $p < 0.01$: significant differences after 72 h observed between 125 and 250 µg/mL aKG, between 250 and 375 µg/mL aKG, and between 375 and 500 µg/mL aKG. NCI-H23: $^{\circ}$ $p < 0.05$: significant differences observed between 200, 250, and 375 µg/mL aKG and the control (0 µg/mL aKG) after 24 h; $^{\&}$ $p < 0.05$: significant differences observed between 200 µg/mL aKG and the control (0 µg/mL aKG) after 72 h.

After 48 h incubation, all used aKG concentrations decreased the mitochondrial activity compared to the control (100 ± 3.2%, $n = 5$). The concentration of 125 μg/mL aKG showed a significant reduction in mitochondrial activity to 71.1 ± 3.1%, ($n = 5$; $p < 0.01$), 200 μg/mL aKG showed a significant reduction to 69.3 ± 4.2%, ($n = 5$; $p < 0.01$), 250 μg/mL aKG showed a significant reduction to 63.1 ± 1.7%, ($n = 5$; $p < 0.01$), 375 μg/mL aKG showed a significant reduction to 55.2 ± 1.8%, ($n = 5$; $p < 0.01$), and 500 μg/mL aKG showed a significant reduction to 44.1 ± 1.8% ($n = 5$; $p < 0.01$).

After 72 h incubation with aKG, the same effects were estimated compared to the results after 48 h. The control signal was detected at 100.0 ± 2.9%, ($n = 5$). Significantly lower levels of the mitochondrial activity were measured using 125 μg/mL aKG (76.5 ± 3.2%, $n = 5$; $p < 0.01$), 200 μg/mL aKG (84.2 ± 2.1%, $n = 5$; $p < 0.01$), 250 μg/mL aKG (60.1 ± 2.1%, $n = 5$; $p < 0.01$), 375 μg/mL aKG (78.1 ± 2.1%, $n = 5$; $p < 0.01$), and 500 μg/mL aKG (46.2 ± 2.1%, $n = 5$; $p < 0.01$). The IC50% of aKG was calculated to be around 500 μg/mL aKG regardless of the incubation time.

NCI-H23 cells treated with aKG showed a significant decrease in the mitochondrial activity after 24 h incubation for 200 (84.5 ± 3.1%; $n = 5$; $p < 0.05$), 250 (84.1 ± 4.2%; $n = 5$; $p < 0.05$), and 375 μg/mL (80.9 ± 5.5%; $n = 5$; $p < 0.05$) compared to the control (100.0 ± 8.4%) and after 72 h between the control signal (100.0 ± 2.0%) and 200 μg/mL aKG (83.2 ± 6.9%; $n = 5$; $p < 0.05$).

5-HMF: Figure 5A shows the influence on mitochondrial activity of MTC-SK cells in the absence or presence of several 5-HMF concentrations after 24, 48, and 72 h. After 24 h using 125 μg/mL 5-HMF, the mitochondrial activity was significantly reduced to 85.2 ± 4.2% ($n = 5$; $p < 0.01$) compared to the control (100.0 ± 4.2%; $n = 5$). There was a stabilization at the same level of the effect with 125 μg/mL 5-HMF as when using 200 μg/mL 5-HMF (84.2 ± 2.5%; $n = 5$), 250 μg/mL 5-HMF (83.2 ± 4.1%; $n = 5$), and 375 μg/mL 5-HMF (85.2 ± 6.8%; $n = 5$). The usage of 500 μg/mL 5-HMF showed a significant decrease in the mitochondrial activity to 70.2 ± 9.8% ($n = 5$; $p < 0.05$), compared to 200 μg/mL 5-HMF.

A 48 h incubation of MTC-SK cells with various 5-HMF concentrations generated a higher effect on the mitochondrial activity compared to 24 h. The highest significant decrease was measured using 500 μg/mL 5-HMF (22.1 ± 2.7%; $n = 5$; $p < 0.01$), compared to the control (100.0 ± 2.8%; $n = 5$), followed by 375 μg/mL 5-HMF (32.1 ± 1.8%; $n = 5$; $p < 0.01$), 250 μg/mL 5-HMF (40.8 ± 2.2%; $n = 5$; $p < 0.01$), 200 μg/mL 5-HMF (46.2 ± 4.1%; $n = 5$; $p < 0.01$), and 125 μg/mL 5-HMF (62.2 ± 4.2%; $n = 5$; $p < 0.01$).

After 72 h incubation of different 5-HMF concentrations, the inhibitory effect on mitochondrial activity was slightly better compared to 48 h incubation. A nearly 80% reduction using 5-HMF was obtained using 500 μg/mL 5-HMF (20.1 ± 0.9%; $n = 5$; $p < 0.01$) compared to the control (100.0 ± 2.9%; $n = 5$), followed by 375 μg/mL 5-HMF (25.7 ± 2.1%; $n = 5$; $p < 0.01$), 250 μg/mL 5-HMF (35.2 ± 1.7%; $n = 5$; $p < 0.01$), 200 μg/mL 5-HMF (52.1 ± 1.8%; $n = 5$; $p < 0.01$), and the lowest concentration of 125 μg/mL 5-HMF (58.9 ± 2.1%; $n = 5$; $p < 0.01$). The IC50% was calculated for 48 and 72 h incubation to be around 200 μg/mL 5-HMF.

Figure 5B shows the reduction in mitochondrial activities of NCI-H23 cells. After 24 h incubation, no significant reduction was obtained using 125–500 μg/mL 5-HMF compared to control (100 ± 8.2%; $n = 5$). Increasing the incubation time to 48 h 5-HMF reduced the mitochondrial activity significantly between the control (100.0 ± 8.2%; $n = 5$) and 125 (79.2 ± 8.3%; $n = 5$; $p < 0.01$), 200 (78.1 ± 7.9%; $n = 5$; $p < 0.01$), 250 (78.2 ± 7.6%; $n = 5$; $p < 0.01$), 375 (70.1 ± 13.5%; $n = 5$; $p < 0.01$), and 500 μg/mL (68.3 ± 9.2%; $n = 5$; $p < 0.01$). After 72 h, no improvement using several 5-HMF concentrations compared to 48 h incubation was detected. The mitochondrial activity of all used 5-HMF concentrations was reduced significantly compared to the control (100.2 ± 7.6%; $n = 5$; $p < 0.01$): 125 (79.0 ± 7.1%; $n = 5$), 200 (74.2 ± 6.8%; $n = 5$), 250 (73.1 ± 7.6%; $n = 5$), 375 (67.2 ± 8.6%; $n = 5$; $p < 0.05$), and 500 μg/mL (66.8 ± 9.8%; $n = 5$).

Combination of aKG + 5-HMF: Figure 6 shows the influence on mitochondrial activity of MTC-SK cells in absence or presence of several combined aKG + 5-HMF concentrations after 24, 48, and 72 h.

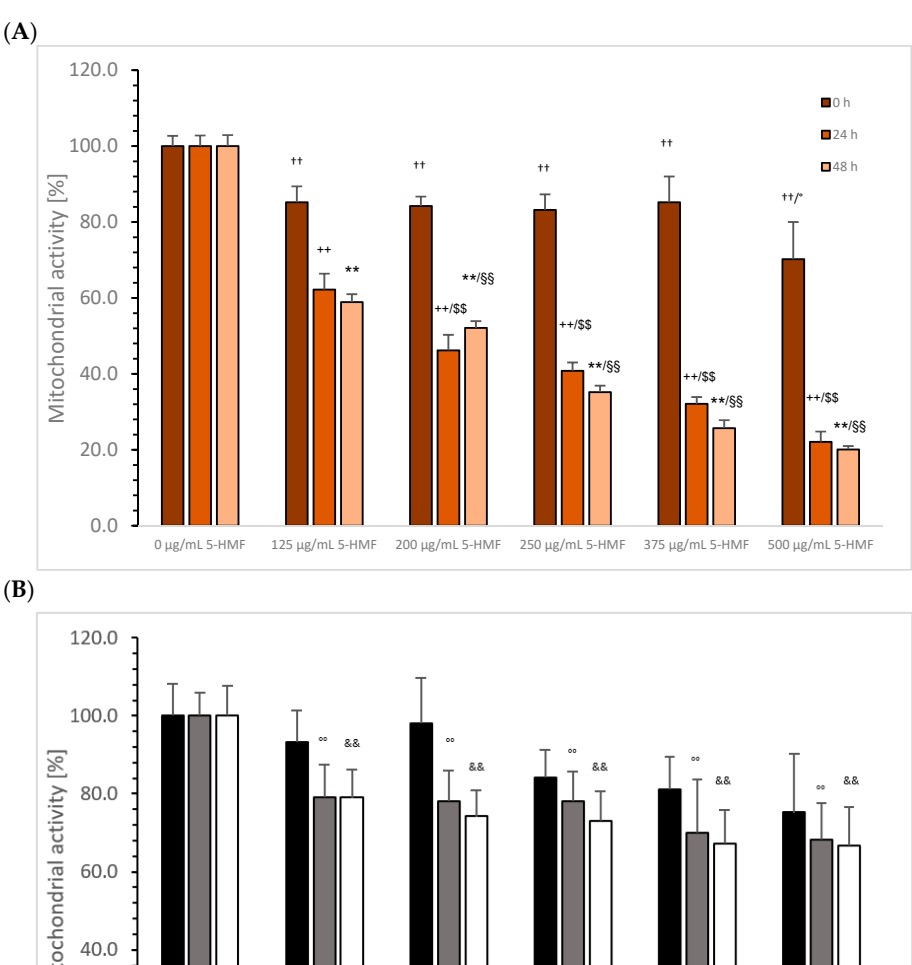

**Figure 5.** Mitochondrial activity of MTC-SK cells (**A**) and NCI-H23 (**B**) in presence or absence of 100, 200, 250, 375, and 500 µg/mL 5-HMF after 24, 48, and 72 h incubation. MTC-SK: $^{\dagger\dagger}$ $p < 0.01$: significant differences after 24 h observed between 100, 200, 250, 375, and 500 µg/mL 5-HMF and the control (0 µg/mL 5-HMF); $^{\circ}$ $p < 0.05$: significant differences after 24 h observed between 500 and 200 µg/mL 5-HMF; $^{++}$ $p < 0.01$: significant differences after 48 h observed between 100, 200, 250, 375, and 500 µg/mL 5-HMF and the control; $^{\$\$}$ $p < 0.01$: significant differences after 48 h observed between 125 and 200 µg/mL 5-HMF, between 250 and 375 µg/mL 5-HMF, and between 375 and 500 µg/mL 5-HMF; ** $p < 0.01$: significant differences after 72 h observed for 100, 200, 250, 375, and 500 µg/mL; $^{\S\S}$ $p < 0.01$: significant differences after 48 h observed between 125 and 200 µg/mL 5-HMF, between 200 and 250 µg/mL 5-HMF, between 250 and 375 µg/mL 5-HMF, and between 375 and 500 µg/mL 5-HMF. NCI-H23: $^{\circ\circ}$ $p < 0.01$: significant differences observed between 0 and 125, 200, 250, 375, and 500 µg/mL 5-HMF after 48 h incubation; $^{\&\&}$ $p < 0.01$; significant differences after 72 h observed between 0 and 125, 200, 250, 375, and 500 µg/mL 5-HMF.

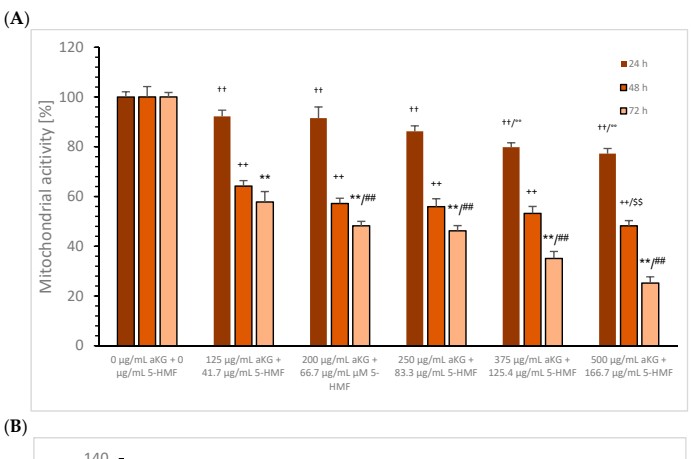

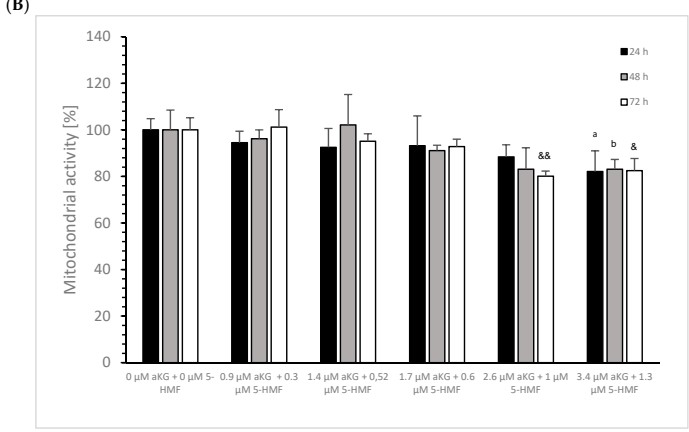

**Figure 6.** Mitochondrial activity of MTC-SK cells (**A**) and NCI-H23 (**B**) in presence or absence of 125 μg/mL aKG + 41.7 μg/mL 5-HMF, 200 μg/mL aKG + 66.7 μg/mL 5-HMF, 250 μg/mL aKG + 83.3 μg/mL 5-HMF, 375 μg/mL aKG + 125.4 μg/mL 5-HMF, and 500 μg/mL aKG + 166.7 μg/mL 5-HMF after 24, 48, and 72 h incubation. MTC-CK: [++] $p < 0.01$: significant differences after 24 h observed for 125 μg/mL aKG + 41.7 μg/mL 5-HMF, 200 μg/mL aKG + 66.7 μg/mL 5-HMF, 250 μg/mL aKG + 83.3 μg/mL 5-HMF, 375 μg/mL aKG + 125.4 μg/mL 5-HMF, and 500 μg/mL aKG + 166.7 μg/mL 5-HMF compared to the control (0 μg/mL aKG + 0 μg/mL 5-HMF); [°°] $p < 0.05$: significant differences after 24 h observed between 250 μg/mL aKG + 83.3 μg/mL 5-HMF and 375 μg/mL aKG + 125.4 μg/mL 5-HMF or 500 μg/mL aKG + 166.7 μg/mL 5-HMF; [++] $p < 0.01$: significant differences after 48 h observed for 125 μg/mL aKG + 41.7 μg/mL 5-HMF, 200 μg/mL aKG + 66.7 μg/mL 5-HMF, 250 μg/mL aKG + 83.3 μg/mL 5-HMF, 375 μg/mL aKG + 125.4 μg/mL 5-HMF, and 500 μg/mL aKG + 166.7 μg/mL 5-HMF compared to the control (0 μg/mL aKG + 0 μg/mL 5-HMF); [$$] $p < 0.01$: significant differences after 48 h observed between 250 μg/mL aKG + 83.3 μg/mL 5-HMF and 500 μg/mL aKG + 166.7 μg/mL 5-HMF; [**] $p < 0.01$: significant differences after 72 h observed for 125 μg/mL aKG + 41.7 μg/mL 5-HMF, 200 μg/mL aKG + 66.7 μg/mL 5-HMF, 250 μg/mL aKG + 83.3 μg/mL 5-HMF, 375 μg/mL aKG + 125.4 μg/mL 5-HMF, and 500 μg/mL aKG + 166.7 μg/mL 5-HMF compared to the control (0 μg/mL aKG + 0 μg/mL 5-HMF); [##] $p < 0.01$: significant differences after 72 h observed between 125 μg/mL aKG + 41.7 μg/mL 5-HMF and 200 μg/mL aKG + 66.7 μg/mL 5-HMF or 250 μg/mL aKG + 83.3 μg/mL 5-HMF, between 250 μg/mL aKG + 83.3 μg/mL 5-HMF and 375 μg/mL aKG + 125.4 μg/mL 5-HMF, and between 375 μg/mL aKG + 125.4 μg/mL 5-HMF and 500 μg/mL aKG + 166.7 μg/mL 5-HMF. NCI-H23: [a] $p < 0.05$: significant differences after 24 h observed between 500 μg/mL aKG + 166.7 μg/mL 5-HMF and the control (0 μg/mL aKG + 0 μg/mL 5-HMF); [b] $p < 0.05$: significant differences after 48 h observed between 500 μg/mL aKG + 166.7 μg/mL 5-HMF and control (0 μg/mL aKG + 0 μg/mL 5-HMF); [&] $p < 0.05$: significant differences after 72 h observed between 500 μg/mL aKG + 166.7 μg/mL 5-HMF and the control (0 μg/mL aKG + 0 μg/mL 5-HMF); [&&] $p < 0.01$: significant differences after 72 h observed between 375 μg/mL aKG + 125.4 μg/mL 5-HMF and the control (0 μg/mL aKG + 0 μg/mL 5-HMF).

Within 24 h incubation, the different aKG + 5-HMF concentrations reduced the mitochondrial activity significantly in a dose-dependent manner: The lowest concentration of 125 µg/mL aKG + 41.7 µg/mL 5-HMF decreased the mitochondrial activity significantly to $92.2 \pm 2.5\%$ ($n = 5$; $p < 0.01$) compared to the control ($100 \pm 2.1\%$ $n = 5$), whereas 200 µg/mL aKG + 66.7 µg/mL 5-HMF decreased it to $91.5 \pm 4.5\%$ ($n = 5$; $p < 0.01$), 250 µg/mL aKG + 83.3 µg/mL 5-HMF decreased it to $86.2 \pm 2.2\%$ ($n = 5$; $p < 0.01$), 375 µg/mL aKG + 125.4 µg/mL 5-HMF decreased it to $79.8 \pm 2.2\%$ ($n = 5$; $p < 0.01$), and 500 µg/mL aKG + 166.7 µg/mL 5-HMF decreased it to $77.2 \pm 2.1\%$ ($n = 5$; $p < 0.01$).

Within 48 h incubation using 125 µg/mL aKG + 41.7 5-HMF, there was a significant reduction in the mitochondrial activity to $64.2 \pm 2.2\%$ ($n = 5$; $p < 0.01$), whereas 200 µg/mL aKG + 66.7 µg/mL 5-HMF decreased it to $57.2 \pm 2.2\%$ ($n = 5$; $p < 0.01$), 250 µg/mL aKG + 83.3 µg/mL 5-HMF decreased it to $55.9 \pm 3.2\%$ ($n = 5$; $p < 0.01$), 375 µg/mL aKG + 125.4 µg/mL 5-HMF decreased it to $53.2 \pm 2.8\%$ ($n = 5$; $p < 0.01$), and 500 µg/mL aKG + 166.7 µg/mL 5-HMF decreased it to $48.2 \pm 2.4\%$ ($n = 5$; $p < 0.01$), which also resulted in an IC50% after 48 h incubation.

After 72 h incubation using the different aKG + 5-HMF combinations, there was a sustained reduction in the mitochondrial activity. A significant reduction in the mitochondrial activity was obtained for 125 µg/mL aKG + 41.7 µg/mL 5-HMF ($57.8 \pm 4.2\%$; $n = 5$; $p < 0.01$), 200 µg/mL aKG + 66.7 µg/mL 5-HMF ($48.2 \pm 1.8\%$; $n = 5$; $p < 0.01$), 250 µg/mL aKG + 83.3 µg/mL 5-HMF ($46.2 \pm 2.1\%$; $n = 5$; $p < 0.01$), 375 µg/mL aKG + 125.4 µg/mL 5-HMF ($40.1 \pm 2.8\%$; $n = 5$; $p < 0.01$), and 500 µg/mL aKG + 166.7 µg/mL 5-HMF ($38.2 \pm 2.5\%$; $n = 5$; $p < 0.01$) compared to the control ($100 \pm 1.8\%$; $n = 5$). The IC50% after 72 h was reached by the addition of 200 µg/mL aKG + 66.7 µg/mL 5-HMF to MTC-SK cells.

### 3.3. Caspase-3 Activity

aKG: After 72 h incubation, MTC-SK cell aliquots were taken for the investigation of caspase-3 activity of the single components aKG or 5-HMF or the combination of both components in different concentrations compared to 4 µM CPT. Figure 7A shows the increase in caspase-3 activity of different aKG concentrations. Only the highest concentration, 500 µg/mL aKG ($21.2 \pm 2.4\%$; $n = 5$; $p < 0.01$), showed a significant increase in caspase-3 activity compared to the control (0 µg/mL aKG). No caspase activity was recorded using 0–500 µg/mL aKG on NCI-H23 cells (Figure 7B).

5-HMF: Figure 7C shows that 125 µg/mL 5-HMF led to a significant increase in the caspase-3 activity compared to the control ($19.8 \pm 1.5\%$ vs. $10.3 \pm 0.9\%$; $n = 5$; $p < 0.01$). Increasing the 5-HMF concentration to 200 µg/mL led to a significantly higher caspase-3 activity of $29.2 \pm 0.8\%$ ($n = 5$; $p < 0.01$) compared to the control, followed by 250 µg/mL 5-HMF, $34.2 \pm 1.9\%$ ($n = 5$; $p < 0.01$), 375 µg/mL 5-HMF, $42.9 \pm 0.8\%$ ($n = 5$; $p < 0.01$), and 500 µg/mL 5-HMF, $59.6 \pm 2.4\%$ ($n = 5$; $p < 0.01$). No caspase activation was measured using all 5-HMF concentrations on NCI-H23 cells (Figure 7D).

The combination of aKG + 5-HMF (Figure 7E): Both substances together increased the caspase-3 activity in a dose-dependent manner. Specifically, 200 µg/mL aKG + 66.7 µg/mL µM 5-HMF led to a significantly higher caspase-3 activity $19.2 \pm 0.8\%$ ($n = 5$; $p < 0.05$) compared to the control ($14.2 \pm 1.9\%$; $n = 5$) after 72 h incubation, followed by 250 µM aKG + 83.3 µg/mL 5-HMF ($22.2 \pm 0.6\%$; $n = 5$; $p < 0.01$), 375 µg/mL µM aKG + 125.7 µg/mL 5-HMF ($42.8 \pm 3.2\%$; $n = 5$; $p < 0.01$), and 500 µg/mL aKG + 1666.7 µg/mL 5-HMF ($55.2 \pm 2.0\%$; $n = 5$; $p < 0.01$).

The usage of several combined aKG + 5-HMF solutions showed no significant effect on NCI-H23 cells (7F).

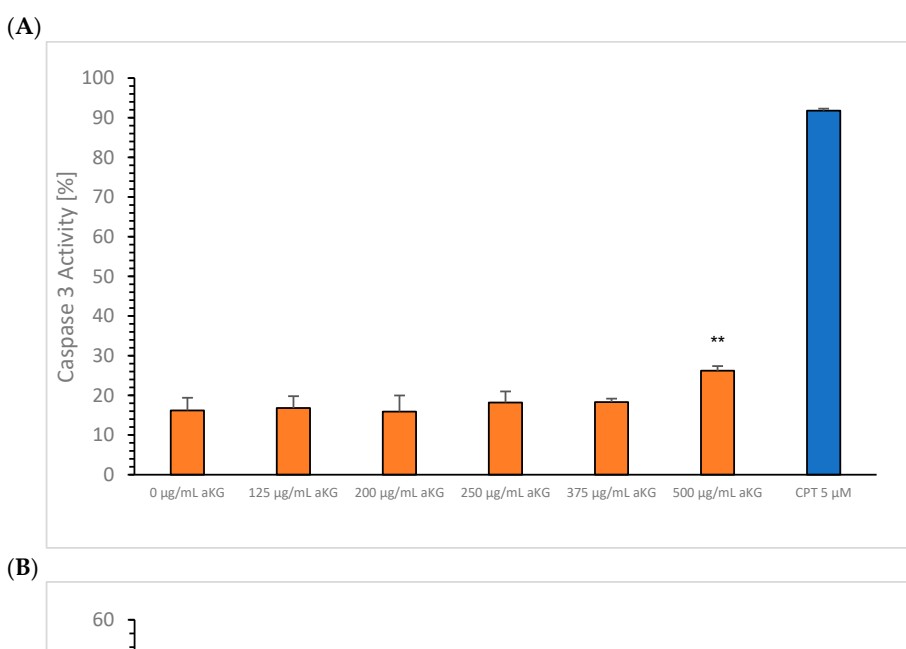

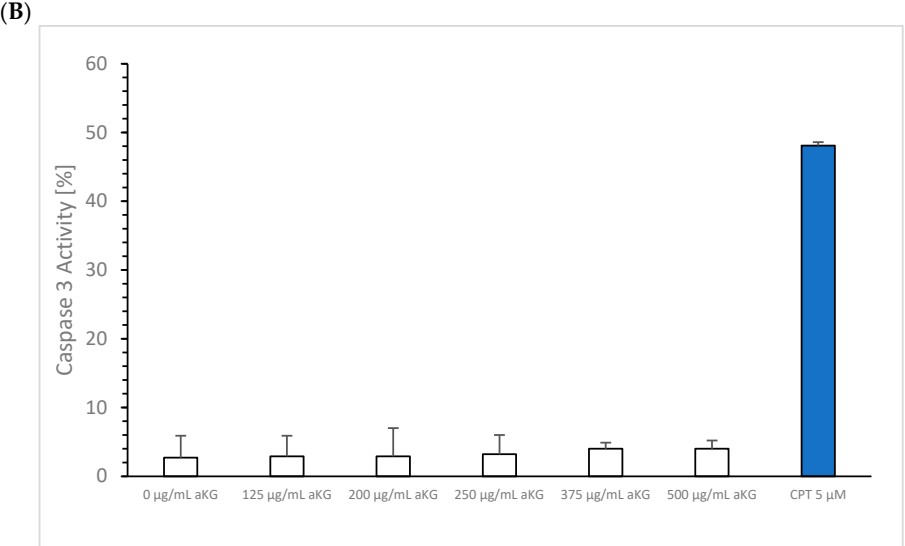

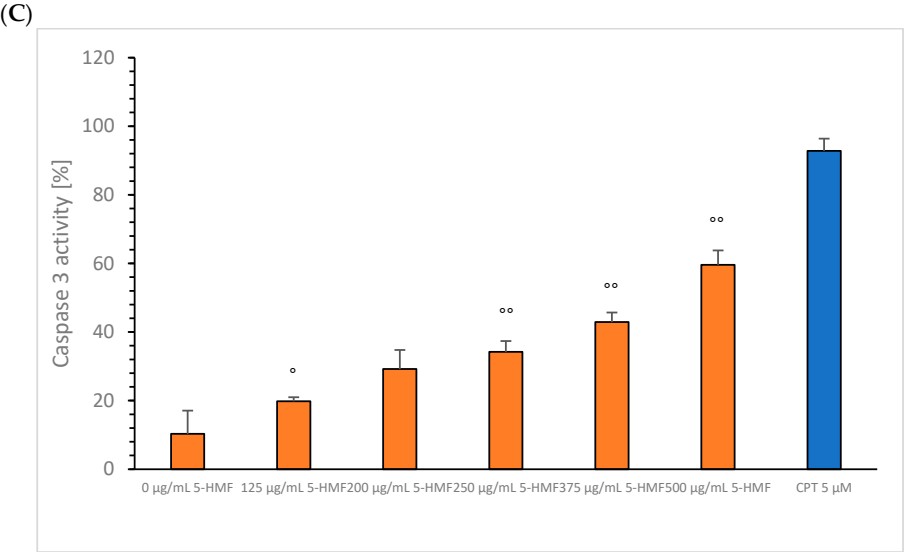

**Figure 7.** *Cont.*

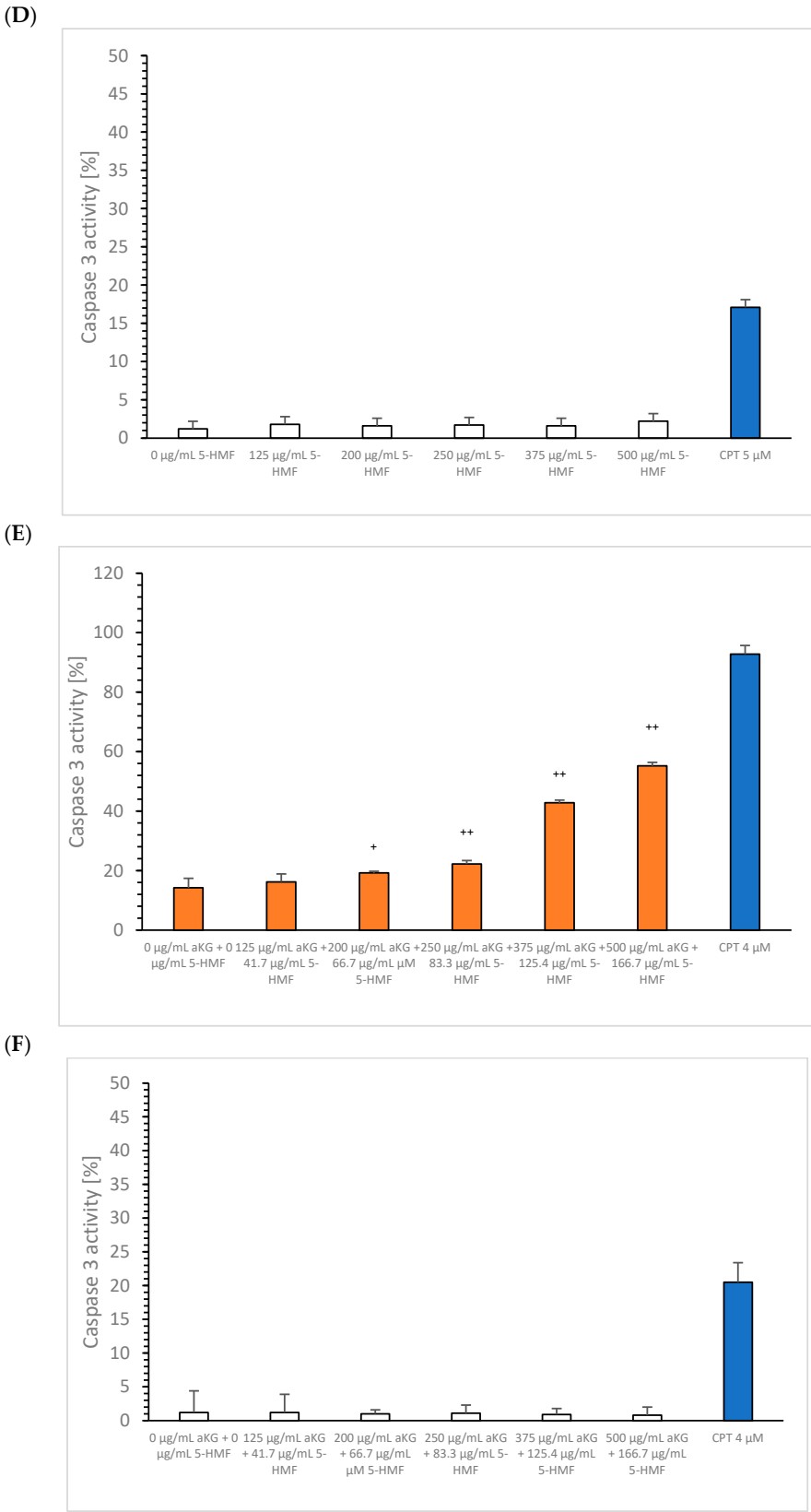

**Figure 7.** Caspase-3 activity after 72 h incubation of the single components 0, 125, 200, 250, 375, and 500 µg/mL aKG (**A**,**B**), 0, 125, 200, 250, 375, and 500 µg/mL 5-HMF (**C**,**D**), and the combinations of 0 µ g/mL aKG + 0 µg/mL 5-HMF, 125 µg/mL aKG + 41.7 µg/mL 5-HMF, 200 µg/mL aKG + 66.7 µg/mL 5-HMF, 250 µg/mL aKG + 83.3 µg/mL 5-HMF, 375 µg/mL aKG + 125.4 µg/mL

5-HMF, and 500 µg/mL aKG + 166.7 µg/mL 5-HMF (**E,F**) compared to 5 µM CPT (blue bars) on MTC-SK (orange bars) or NCI-H23 (white bars). ** $p < 0.01$: significant differences after 72 h observed between 0 and 500 µg/mL aKG. ° $p < 0.05$: significant differences after 72 h observed between 0 and 125 µg/mL 5-HMF; °° $p < 0.01$: significant differences after 72 h observed between 125 and 200 µg/mL 5-HMF, between 250 and 375 µg/mL 5-HMF, and between 375 and 500 µg/mL 5-HMF. + $p < 0.05$: significant differences after 72 h observed between 0 µg/mL aKG + 0 µg/mL 5-HMF and 200 µg/mL aKG + 66.7 µg/mL 5-HMF; ++ $p < 0.01$: significant differences after 72 h observed between 200 µg/mL aKG + 66.7 µg/mL 5-HMF and 250 µg/mL aKG + 83.3 µg/mL 5-HMF, between 250 µg/mL aKG + 83.3 µg/mL 5-HMF and 375 µ g/mL aKG + 125.4 µg/mL 5-HMF, and between 375 µg/mL aKG + 125.4 µg/mL 5-HMF and 500 µg/mL aKG + 166.7 µg/mL 5-HMF.

### 3.4. Carbonylated Membrane Proteins

Figure 8A shows the content of oxidatively modified proteins in the membrane of MTC-SK in presence or absence of different aKG + 5-HMF concentrations after 72 h. Incubation of 250 µg/mL aKG + 83.3 µg/mL 5-HMF reduced the carbonylated protein content significantly to $6.7 \pm 0.3$ nmol/mg ($n = 3$; $p < 0.01$) compared to the control ($8.7 \pm 1.1$ nmol/mg; $n = 3$), followed by 375 µg/mL aKG + 125.4 µg/mL 5-HMF ($5.4 \pm 0.5$ nmol/mg; $n = 3$) and 500 µg/mL aKG + 166.7 µg/mL 5-HMF ($4.2 \pm 0.9$ nmol/mg; $n = 3$). The usage of several combined aKG + 5-HMF solutions showed no significant effect on NCI-H23 cells (Figure 8B).

Figure 9 shows the linear regression between the mitochondrial activity of the MTC-SK cells and the content of membrane carbonylated proteins in the absence or presence of aKG + 5-HMF after 72 h incubation. The regression term was calculated with $r^2 = 1$. The highest concentration (500 µg/mL aKG + 166.7 µg/mL µM 5-HMF) showed the lowest protein modification per mg cell membrane protein and the lowest mitochondrial activity, whereas the lowest concentration (125 µg/mL aKG + 41.7 µg/mL 5-HMF) showed a high oxidative modified protein level with a high cell growth. The mitochondrial activity of MTC-SK in absence of any substance was 2.5-fold lower compared to the highest used aKG + 5-HMF solution.

**(A)**

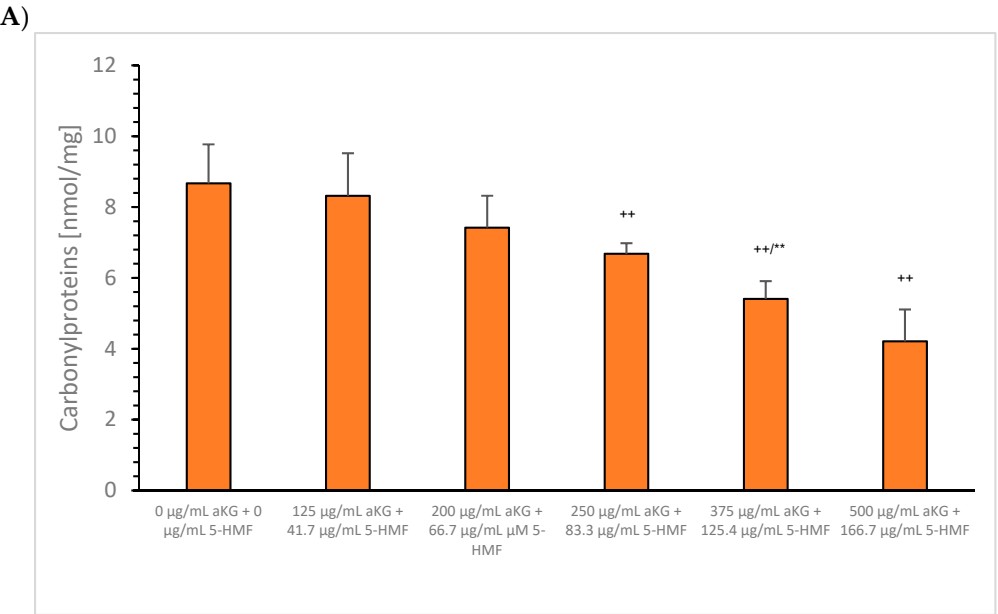

**Figure 8.** *Cont.*

**(B)**

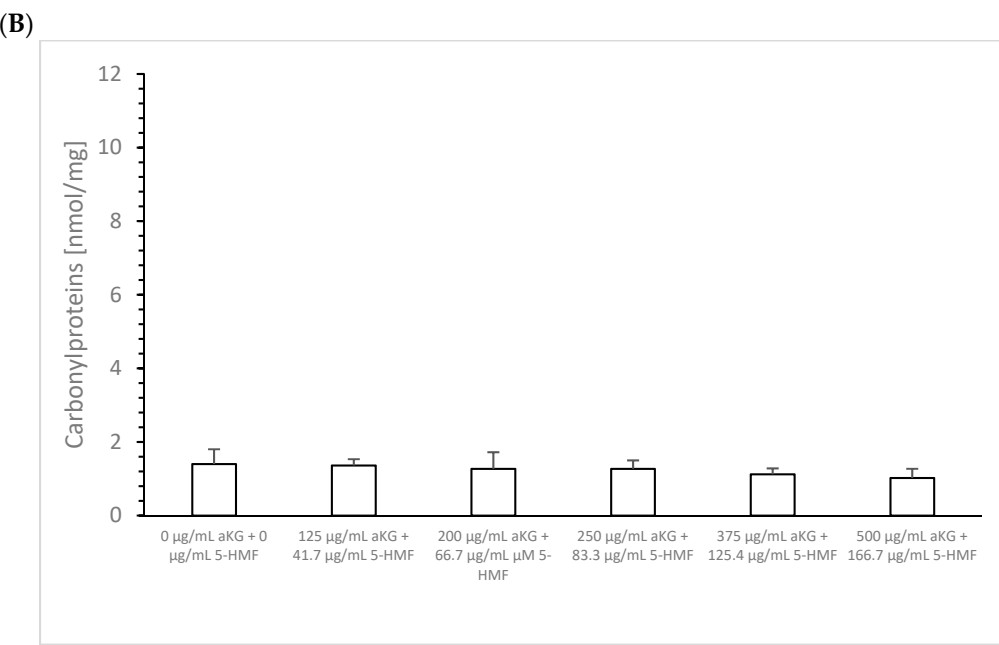

**Figure 8.** Generated oxidatively modified membrane proteins on MTC-SK (**A**) and NCI-H23 (**B**) cells, namely, carbonylated proteins, with several aKG + 5-HMF concentrations after 72 h incubation. [++] $p < 0.01$: significant differences after 72 h observed for the control compared to 1.7 μM aKG + 0.6 μM 5-HMF, 2.6 μM aKG + 1 μM 5-HMF, and 3.4 μM aKG + 1.3 μM 5-HMF; [**] $p < 0.01$: significant differences after 72 h observed between 1.7 μM aKG + 0.6 μM 5-HMF and 2.6 μM aKG + 1 μM 5-HMF.

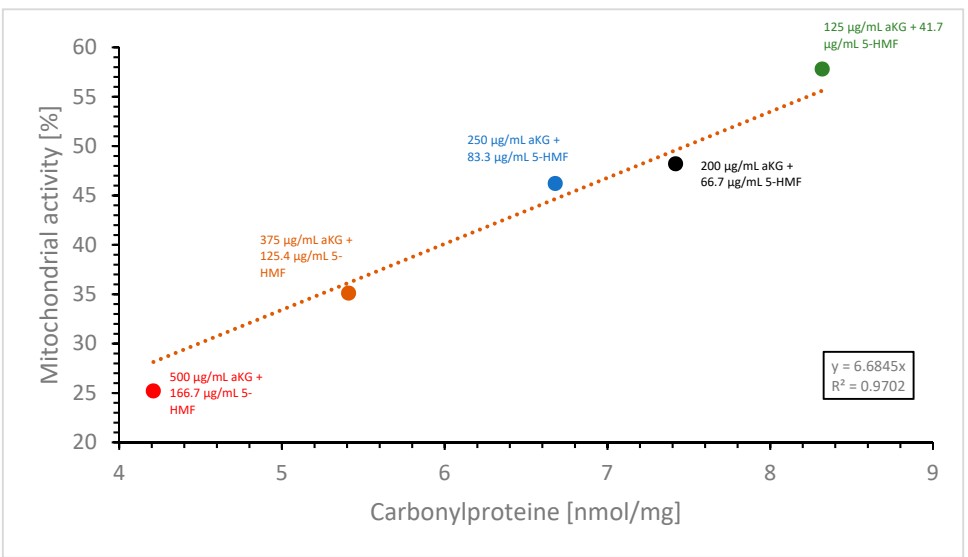

**Figure 9.** Correlation between the content of carbonylated membrane proteins as a marker of oxidative stress and the mitochondrial activity of MTC-SK cells in presence of different aKG + 5-HMF concentrations after 72 h incubation.

## 4. Discussion

Medullary thyroid carcinoma (MTC) is a rare neuroendocrine tumor that originates from parafollicular cells [17]. It accounts for 5–10% of all thyroid cancers. MTC can occur sporadically or be inherited, with the familial form present in 25% of patients [18]. Unlike well-differentiated thyroid cancers, MTC requires distinct diagnostic and therapeutic

approaches. Recently, molecular diagnosis and individualized treatment have gained importance for researchers and clinicians [17].

Familial MTC syndromes include multiple endocrine neoplasia (MEN) types 2A and 2B, and familial MTC, which are inherited in an autosomal dominant manner. Children with any of these syndromes have a 100% risk of developing MTC [19]. Therefore, genetic testing is recommended for the diagnosis of familial MTC syndromes. In MEN 2A, cysteine residue substitutions in exons 10 and 11 are observed, while threonine-for-methionine substitution in codon 918 of exon 16 is seen in MEN 2B. In familial MTC, mutations occur in exons 10, 13, and 14 [20].

The updated guidelines recommend ultrasound evaluation of thyroid nodules and measurement of serum thyroid-stimulating hormone (TSH) for diagnosing MTC. On the basis of the results, particularly the nodule size, a fine needle aspiration biopsy (FNAB) is advised [20]. In some cases, additional serum calcitonin and radionuclide imaging may be recommended. If the FNAB results confirm MTC, an ultrasound of the neck, serum calcitonin assay, serum carcinoembryonic antigen (CEA) measurement, and RET germline mutation analysis should be performed [21].

In the case of incomplete tumor resections, external beam radiation therapy can be considered (EBRT) [22].

For MTC patients with a tumor size $\geq 1$ cm$^3$, total thyroidectomy and bilateral central neck dissection are recommended. In cases where MTC tumors are $\leq 1$ cm, total thyroidectomy is sufficient. Consideration of total neck dissection depends on calcitonin levels higher than 200 pg/mL [20]. External beam radiation therapy (EBRT) can be considered in cases of incomplete tumor resections [21].

In metastatic MTC, vandetanib and cabozantinib are approved by regulatory agencies for administration. The National Comprehensive Cancer Network (NCCN) recommends involving patients with symptomatic or progressing MTC in clinical trials with targeted therapies, immunotherapy, small-molecule kinase inhibitors, or decarbazine (DTIC)-based chemotherapy with EBRT. Biophosphonate and denosumab treatment is recommended for bone metastases [23].

While surgery remains the standard treatment for MTC, there is growing interest in using medication such as tyrosine kinase inhibitors (TKIs), e.g., vandetanib, cabozantinib, selpercatinib, and pralsetinib [20]. These TKIs target proteins such as MET, RET, VEGFR-2, and EGFR, playing important roles in MTC tumorigenesis. Cabozantinib targets the RET, MET, VEGFR-1,-2,-3, KIT, TrkB, FLT-3, ACL, and TIE-2 pathways. Selpercatinib is recommended for RET-mutant MTC, while pralsetinib selectively targets RET alterations.

In the context of RET-mutant medullary thyroid carcinoma (MTC), it is recommended to use selpercatinib as a kinase inhibitor targeting various forms of rearranged during transfection (RET), including wildtype RET, multiple mutated RET isoforms, VEGFR1, and VEGFR3. Pralsetinib has shown selective activity against RET alterations [24].

Preclinical studies suggest the involvement of not only RET but also VEGFR, EGFR, and MET in the tumorigenesis of MTC [25].

In this study, we investigated the effects of alpha-ketoglutarate (aKG) and 5-hydroxy methylfurfural (5-HMF) on MTC cells, focusing on their antioxidative, antiproliferative, and anticarcinogenic properties.

The individual compound aKG is primarily involved in the Krebs cycle to prevent hypoxia and processes such as the activation of HIF$\alpha$, upregulation of transcription genes, and neo-angiogenesis, which play crucial roles in the cancerogenesis of solid tumors [5]. HIF-1, under hypoxic conditions in tumor cells, activates key components such as VEGF, PDGF-B (platelet-derived growth factor), hepatocyte growth factor, epidermal growth factor, angiopoietin-2, and placental growth factors, due to reduced levels of aKG and increased onco-metabolites such as succinate and fumarate. Therefore, aKG serves as an important antiangiogenic agent in tumors, as confirmed by Matsumoto et al. [26], who administered exogenous aKG to two tumor cell lines in vitro. aKG, either alone or in combination with 5-FU, significantly reduced the size of transplanted tumors in mice.

Reduced aKG levels can also explain the increased HIF-1 activity and proliferation of tumor cells under normoxia. This phenomenon, known as pseudohypoxia, was tested by Briere et al. [27], where aKG inhibited the translocation of HIF-1 in fibroblasts with succinate dehydrogenase subunit A (SDHA) mutation. Furthermore, exogenously administered aKG [28] could reverse the inhibition of PDH, a component of the HIF-1 pathway, caused by succinate and fumarate.

Further investigations using radiolabeled carbon-marked aKG showed delayed cell penetration. However, the use of derivatized (esters) and radiolabeled aKG demonstrated better penetration into the cells. Nonetheless, the effect of inhibiting HIF-1 in mutated SDHA fibroblasts was lower with derivatized aKG compared to nonderivatized aKG. Exogenously administered aKG to tumor cells showed a highly dose-dependent reduction in cell growth, mitochondrial activity, and caspase activation in osteosarcoma cells [9], colon adenocarcinoma cells [29], and hematopoietic cells [10], indicating the effective cellular penetration of aKG. This can be explained by the following properties of aKG:

(i) Decarboxylation of carbon dioxide (labeled with $^{14}$C): Long et al. [30] revealed that culture medium, in the presence or absence of normal cells, generate hydrogen peroxide. Exogenously administered aKG can react with hydrogen peroxide to generate carbon dioxide and succinate. However, the radioactivity of $^{14}$C-labeled aKG was washed out after cell separation before measurement, resulting in a lack of radioactivity signal. Recent research has also shown that aKG reacts with peroxynitrite (ONOOH) to generate succinate and carbon dioxide at physiological pH in media and cells [31].

(ii) pH in medium: No chemical reaction of ONOOH or $H_2O_2$ can be detected at a pH of 6. An increase in incubation time leads to increased acidity of the medium. Therefore, aKG must be stable in the presence of peroxides and able to penetrate cells. Administering aKG to human fibroblasts did not lead to an increase in cell growth but increased mitochondrial activity, indicating unhindered penetration of aKG into human cells [31].

(iii) Tumor cells: Due to elevated turnover rates for energy generation, the pH conditions in tumor cells are more acidic (lactate metabolism) compared to normal cells. Under these conditions, aKG is more stable and can effectively penetrate tumoral cells and their mitochondria.

(iv) Increasing incubation time results in a decrease in pH. While some studies have shown the antitumoral effect of aKG using only a single incubation time point, we investigated the effects after 24, 48, and 72 h. The best effect of exogenously administered aKG on inhibiting cell growth and mitochondrial activity in MTC-SK cells was observed after 72 h using the highest concentration of aKG [10]. Similar results were observed in leukemic cells. Furthermore, 500 μg/mL of aKG did not influence cell growth but increased mitochondrial activity in normal human fibroblasts, suggesting that aKG, despite penetrating cells and mitochondria, does not play a role in the proliferation of normal cells.

(v) Oxidative stress: It is widely accepted that cancer cells generate more radicals (including free radicals and reactive oxygen and nitrogen species) compared to normal healthy cells. Detecting oxidative stress in cultured cells is challenging due to the stability and reactivity of these species. Most radicals interact with various organic substances, including monomeric (amino acids, fatty acids, and carbohydrates) or polymeric (proteins, lipids, and di- and polysaccharides) components, leading to modified substances. However, aKG is one of the few organic substances that decomposes to succinate and carbon dioxide in the presence of radicals without prior modification. Glucose, for example, converts to malondialdehyde, polyunsaturated fatty acids produce free carbonyls (MDA, 4-hydroxynonenal), tyrosine results in nitro-tyrosine, and guanosine forms the mutagenic compound 2-hydroxy-guanosine in the presence of radicals. Compared to NADH$^+$H$^+$ (−0.32 volts), FADH$_2$ (−0.22 volts), ubiquinone (0.06 volts), or succinate (0.3 volts), aKG has a higher reduction capacity (−0.38 volts). Carbonylated proteins (CP) have been established as reliable markers for determining oxidative stress levels in cells. The isolated CP content in membranes of MTC-SK is 3–4 times higher compared to unaffected fibroblasts,

and similar results have been observed in Jurkat cells [10], but to a lesser extent in NCI-H23 cells. It appears that antioxidative substances such as aKG and 5-HMF are not able to increase cell death in NCI-H23 cells due to low radical formation expressed in CP. Conversely, pro-oxidative mechanisms, such as vincenin-2, a pro-oxidant flavonoid, increase radical formation, as identified by increased CP, and lead NCI-H23 cells to undergo apoptosis.

In conclusion, aKG is a potent antioxidative substance that counteracts the cell growth of cancer cells through its multifunctional reactions, particularly involving the HIF-1 mechanism. 5-HMF also possesses antioxidative potential, an upregulating effect on antioxidative enzymes, an inhibiting effect on free radicals (due to its structural similarity to vitamin C), and an antitumoral effect [32,33]. However, unlike aKG, 5-HMF is not involved in any metabolic or enzymatic reactions. It is unstable against oxidation or sulfurization (e.g., bacterial infection), resulting in the formation of the mutagenic and genotoxic substance 5-SMF. When administered in combination, aKG was combined with 5-HMF in a ratio of 3:1 to utilize their synergistic potential against oxidative stress and nitration while avoiding any modification of 5-HMF into the potentially toxic 5-sulfoxymethylfurfural (5-SMF).

Although 5-HMF has a decreased ability to reduce reactive oxygen and nitrogen species (RONS), it is more effective than aKG in preventing the nitration of tyrosine residues on proteins. Additionally, 5-HMF upregulates antioxidative acting enzymes and has an inhibiting effect on free radicals [33]. Therefore, since free radicals are necessary for several induction and signaling pathways, especially in the immune system, the administration of higher doses of 5-HMF may be counterproductive in therapeutic aspects.

Both aKG and 5-HMF effectively reduce the content of CP in isolated membrane proteins in a dose-dependent manner, indicating that energy metabolism-generated oxidative stress and disrupted oxidative turnover lead to apoptosis in cancer cells, as shown in Figure 10.

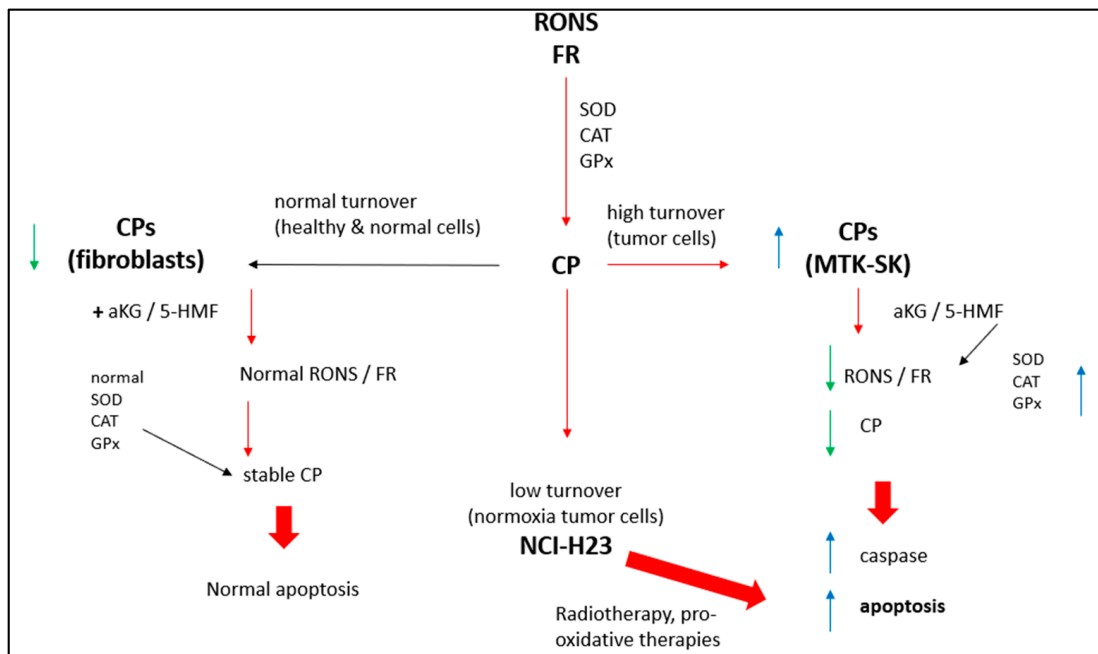

**Figure 10.** RONS and apoptosis: red arrows = reaction; green arrows = decreasing parameters; blue arrows = increasing parameters. CPs in fibroblasts are low because of a normal turnover. Addition of aKG + 5-HMF does have no influence on apoptosis. CPs in NCI-H23 are equal low as like as fibroblasts. No effect was obtained with the addition of aKG + 5-HMF. Only radio- or pro-oxidative therapies lead NCI-H23 cells to apoptosis. High CP levels in MTK-SK cells presents high turnover tumor cells. Addition of 5-HMF + aKG lead to higher enzymatic activities (SOD, CAT, GPx), reduction of free radicals and RONS, induction of caspase activity and apoptosis.

## 5. Conclusions

The antioxidative, antiproliferative, and anticancerogenic effects of aKG + 5-HMF, as previously administered to cancer patients, were confirmed in MTC-SK cells, as well as in previous studies on leukemic cells. Further studies are needed in cell lines with high proliferating properties and high energy rates to confirm these modes of action. Hopefully, future clinical trials in cancer patients will verify the remarkable results observed in in vitro cell studies, such as those conducted in non-small-cell lung cancer patients, and explore the potential of aKG + 5-HMF as an additive metabolic therapy in cancer treatment.

**Author Contributions:** J.G., R.W. and R.H. conceived and designed the experiments. J.G. and P.S. performed the experiments. J.G., R.W. and R.H. analyzed data and J.G. and K.E. wrote the manuscript. J.G., R.W. and P.S. contributed reagents/materials/analysis tools. All authors have read and agreed to the published version of the manuscript.

**Funding:** This research received no external funding.

**Institutional Review Board Statement:** Not applicable.

**Informed Consent Statement:** Not applicable.

**Data Availability Statement:** The data presented in this study are available in article.

**Conflicts of Interest:** The authors declare no conflict of interest.

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
