# Peer review of "Different RONS Generation in MTC-SK and NSCL Cells Lead to Varying Antitumoral Effects of Alpha-Ketoglutarate + 5-HMF"

_cimb, doi:10.3390/cimb45080410_

Round 1

Reviewer 1 Report

Comments and Suggestions for Authors

The research article on “Alpha-ketoglutarate and 5-hydroxymethylfurfural induce 2 apoptosis in medullary cancer cells, while exhibiting reduced 3 efficacy in non-small lung carcinoma cells: an investigation 4 into mitochondrial reactive oxygen and nitrogen species gener- 5 ation” is an interesting article with lots of data.

The arrangement of data and the information regarding how ROS and the effect of individual and combination of mitochondria-targeting antioxidants in MTC-SK or NCI-H23 cells is a bit difficult to follow. The manuscript requires major revisions with an arrangement of data and findings in a better way.

ROS generation is a continuous process and it can increase with chemotherapy. Evaluation of data over 3 days could be very late, especially if the media is not changed for 3 days, addition of agents, ROS and other metabolic byproducts may impact the observations. It is not clear why just 3 is selected vs effector caspases 3/7. Also it is not clear, if the caspase 3 levels were normalized to cell number.

Other limitations are just limiting to analyzing CPs and not looking at other markers such as lipid peroxidation etc., ignoring the fate of the activation of anti-oxidant defense mechanisms.  

Author Response

Lieber Rev. 1,

Vielen Dank für Ihre Kommentare. Wir haben fast alle Ihre Punkte berücksichtigt.

Wir haben versucht, den Ergebnisteil klarer zu gestalten, z. B. die MTC-Ergebnisse in Abbildung 1 und die NCI-Ergebnisse in Abbildung 2 zusammenzustellen. Aber dann haben wir 9-10 Zahlen und die Legenden einschließlich der statistischen Beschreibung sind zu lang und schwieriger zu lesen. Also behalten wir es auf die gleiche Weise bei.

Weitere Punkte sind in der beigefügten Datei beschrieben.

Reviewer 2 Report

Comments and Suggestions for Authors

The manuscript is very interesting. The methodology used is appropriate and sufficient. The manuscript is well written. However, I do have some comments.

I. Major comments:

1. I suggest including a brief paragraph on the role of oxidative stress and nitrosative stress in mitochondrial dysfunction.

2. It is necessary to discuss potential molecular pathways that would allow us to understand how the intervention influences apoptosis. Especially as mitochondrial dysfunction could influence cell death.

3. It would be necessary to include a figure that presents the main molecular pathways involved in the effects.

II. Minor comments:

1. The title is very long, I suggest reducing it

2. Improve the wording of the study objective

Comments on the Quality of English Language

The manuscript is well written, but some editorial changes need to be corrected.

Author Response

Liebe Rev. 2,

Vielen Dank für Ihre Kommentare, die unsere Ergebnisse in unserem Manuskript verbessern.

In der beigefügten Datei haben wir Haupt- und Nebenpunkte Ihrer Bewertung herausgearbeitet.

Mit freundlichen Grüßen

Greilberger

Round 2

Reviewer 2 Report

Comments and Suggestions for Authors

Authors answered all my comments.